**DRIFTS band areas as measured pool size proxy to reduce parameter uncertainty of soil organic matter models**

Moritz Laub[1], Michael Scott Demyan[2], Yvonne Funkuin Nkwain[1], Sergey Blagodatsky[1,3],

Thomas Kätterer[4], Hans-Peter Piepho[5], Georg Cadisch [1]

5   [1] Institute of Agricultural Sciences in the Tropics (Hans-Ruthenberg-Institute), University of Hohenheim, 70599 Stuttgart, Garbenstrasse 13, Germany

[2] School of Environment and Natural Resources, The Ohio State University, Columbus, 2021 Coffey Rd., OH, USA, 43210

[3] Institute of Physicochemical and Biological Problems in Soil Science, Russian Academy of Sciences, 142290 10   Pushchino, Russia

[4] Department of Ecology, Swedish University of Agricultural Sciences, Uppsala, Ulls Väg 16, Sweden

[5] Institute of Biostatistics, University of Hohenheim, 70599 Stuttgart, Fruwirthstr. 23, Germany

*Correspondence to*: Moritz Laub (moritz.laub@uni-hohenheim.de) and Georg Cadisch (georg.cadisch@uni-hohenheim.de)

*Abbreviations: Diffuse reflectance mid infrared Fourier transform spectroscopy (DRIFTS), DRIFTS stability index (DSI) ,equal wight (EW), original weight (OW), soil microbial biomass (SMB), squared model error (SME),*
*soil organic carbon (SOC), soil organic matter (SOM)*

**Abstract.** Soil organic matter (SOM) turnover models predict changes in SOM due to management and environmental factors. Their initialization remains challenging as partitioning of SOM into different hypothetical pools is intrinsically linked to model assumptions. Diffuse reflectance mid infrared Fourier transform spectroscopy
(DRIFTS) provides information on SOM quality and could yield a measurable pool partitioning proxy for SOM. This study tested DRIFTS derived SOM pool partitioning using the DAISY model. The DRIFTS stability index (DSI) of bulk soil samples was defined as the ratio of the area below the aliphatic absorption band (2930 cm$^{-1}$) to the area below the aromatic/carboxylate absorption band (1620 cm$^{-1}$). For pool partitioning, the DSI (= 2930 cm$^{-1}$/1620 cm$^{-1}$) was set equal to the ratio of fast/slow cycling SOM. Performance was tested by simulating
long-term bare fallow plots from the "Bad Lauchstädt extreme farmyard manure experiment" in Germany (Chernozem, 25 years), the "Ultuna continuous soil organic matter field experiment" in Sweden (Cambisol, 50 years) and 7 year duration bare fallow plots from the Kraichgau and Swabian Jura regions in Southwest Germany (Luvisols). All experiments were agricultural fields for centuries before fallow establishment, so classical theory would suggest that steady state can be assumed for initializing SOM pools. Hence, steady state and DSI
initializations were compared, using two published parameter sets, differing in turnover rates and humification efficiency. Initialization using DSI significantly reduced DAISY model error for total soil organic carbon and microbial carbon for cases where assuming steady state had poor model performance. This was irrespective of the

parameter set, but faster turnover performed better for all sites except for Bad Lauchstädt. These results suggest that soils, although under long-term agricultural use, were not necessarily at steady state. In a next step, Bayesian calibration inferred best-fitting turnover rates for DAISY using the DSI were evaluated for each individual site or for all sites combined. Two approaches significantly reduced parameter uncertainty and equifinality in Bayesian calibrations: 1) adding physicochemical meaning with the DSI (for humification efficiency and slow SOM turnover), and 2) combining all sites (for all parameters). Individual site derived turnover rates were strongly site specific. The Bayesian calibration combining all sites suggested a potential for rapid SOM loss with 95 % credibility intervals for the slow SOM pools' half-life being 278 to 1095 years (highest probability density at 426 years). The credibility intervals of this study were consistent with several recently published Bayesian calibrations of similar two-pool SOM models, i.e. turnover rates being faster than earlier model calibrations suggested, hence they likely underestimated potential SOM losses.

## 1    Introduction

Process-based models of plant-soil ecosystems are used from plot to global scales as tools of research and to support policy decisions (Campbell and Paustian, 2015). In soil organic matter (SOM) models, SOM is traditionally divided into several pools, representing fast and slow cycling or even inert SOM (Hansen et al., 1990; Parton et al., 1993). However, these theoretical SOM pools cannot easily be linked to measurable fractions. As a workaround, common methods of SOM pool initialization require that one assumes SOM at steady state or includes a model spin-up run, attempting to simulate SOM dynamics according to history and carbon inputs for the decades to several millennia prior to the period of actual interest (e.g. O'Leary et al., 2016). Theoretically if SOM pools are at steady state and turnover times of SOM pools are known, models could be initialized, i.e. pool sizes calculated, either by simple equations (e.g. for DAISY, Bruun and Jensen, 2002) or by inverse modeling (for RothC, Coleman and Jenkinson, 1996). In most cases, data is insufficient to guarantee that the assumptions of SOM steady state or long-term land use history and inputs are correct, given the lack of data of residue/manure input and weather variability for the required long-term timescales (> 200 years to millennia). Furthermore, exact turnover times of different SOM pools are unknown, which makes the results of inverse modelling and steady state initializations a direct result of model assumptions (Bruun and Jensen, 2002). Hence, it is critical to find measurable proxies such as soil size density fractionation or infrared spectra (Sohi et al., 2001), that can provide information on the quality of SOM and help to disconnect the intrinsic link between turnover times and SOM pool division for SOM pool initialization.

As was shown by Zimmermann et al. (2007), and recently confirmed by Herbst et al. (2018), a link exists between soil fractions obtained by size/density fractionation and fast and slow cycling SOM pools. However, Poeplau et al. (2013) showed, that the same fractionation protocol led to considerably different results at six different laboratories which regularly applied the technique (coefficient of variation from 14 to 138 %). The resulting differences in the model initializations for simulated SOM loss after 40 years of fallow, led to differences in SOM losses that were to up to 30 % of initial SOM. Hence there is a need for a reproducible proxy for SOM pool initialization to reduce the high uncertainty of SOM models. We hypothesized that such a proxy could be obtained from inexpensive, high-throughput Diffuse Reflectance mid Infrared Fourier Transform Spectroscopy (DRIFTS).

As a novel approach, this study uses information gained from DRIFTS spectra to partition measured SOM into pools of different complexity. DRIFTS can provide information on SOM quality, but also on texture and even mineralogy (Nocita et al., 2015; Tinti et al., 2015). The absorbance of mid-infrared light by molecular bonds in the soil sample vibrating at the same frequency produces typical absorption bands at distinct wavelengths (Stevenson, 1994). The area below absorption bands (in short – band area), can be linked to different molecular

bonds of carbohydrates, amides, silicates and others. Two important absorption bands that provide information on SOM quality are the aliphatic carbon band (in short 2930 $cm^{-1}$, limits: 3010 - 2800 $cm^{-1}$) and the aromatic/carboxylate band (in short 1620 $cm^{-1}$, limits: 1660 – 1580 $cm^{-1}$) (Giacometti et al., 2013; Margenot et al., 2015; Pengerud et al., 2013). While both bands are subject to interference (2930 $cm^{-1}$ mainly from water and 1620 $cm^{-1}$ mainly from minerals (Nguyen et al., 1991)), it should be possible to limit the interference using subregions

of the absorption bands with carefully selected integration limits. Indeed, Demyan et al. (2012) found aliphatic carbon to be enriched under long-term farmyard manure application and depleted in mineral fertilizer or control treatments, and showed that the ratio of the 1620 $cm^{-1}$ to 2930 $cm^{-1}$ band area had a significant positive correlation with the ratio of stable to labile SOM obtained by size and density fractionation. It was further corroborated that the band area they used, which mainly selected the top subregion of the absorption bands, are strongly reduced or

lost during combustion (Demyan et al., 2013). Hence, we hypothesized that the ratio of areas below aliphatic to aromatic/carboxylate carbon absorption bands can be used as proxy for the ratio of fast to slow cycling SOM for pool initialization, thus providing a major improvement over assuming steady state SOM. The ratio of areas below absorbance bands of aliphatic to aromatic/carboxylate carbon, will be referred to as the DRIFTS stability index (DSI) hereafter. Testing, improvement and proper use of the DSI was the central topic of this study. Recent findings

have highlighted that the residual water content in bulk soil samples after drying at different temperatures affects the DSI considerably. Water absorbance affects significant parts of the mid-infrared spectra and particularly influences the 2930 and 1620 $cm^{-1}$ band areas (Laub et al., 2019). For this reason, we also tested how the drying temperature prior to DRIFTS measurements affects the use of the DSI proxy, using 32, 65 and 105°C as pretreatment temperatures.

To test our hypotheses about DSI performance, we used the DAISY SOM model (Hansen et al., 2012). DAISY is a commonly used SOM model (Campbell and Paustian, 2015) with a typical multi-pool structure, which includes two soil microbial biomass (SMB) pools, as well as two pools for stabilized SOM (fast and slow cycling). With first-order turnover kinetics and a humification efficiency parameter (**Figure 1**), the DAISY structure is similar to other widely used SOM models such as CENTURY (Parton et al., 1993) or ICBM (Andrén and Kätterer, 1997).

Model SOM pool initialization using the DSI was compared to initialization via a steady state assumption with different published turnover rates. For this comparison bare fallow experiments from a range of different sites and time scales from one to five decades were included. Bare fallow experiments were used to avoid the added complexity caused by the conversion of different plant compounds into SOM of varying stabilities during decomposition.

As SOM pool sizes and turnover rates are closely linked, it could also be necessary to recalibrate DAISY parameters for the use of the DSI. Therefore, a Bayesian calibration of turnover rates was used to adjust DAISY turnover rates to the pool division and time dynamics of the measured DSI throughout the fallow period. Thus, the

DAISY parameterization was evaluated with respect to equifinality and uncertainty as well as dependence on model structure. The final hypothesis was, that through a Bayesian calibration using the DSI, DAISY pools will

correspond to measured, i.e. physiochemically meaningful fractions thus reducing uncertainty. The posterior credibility intervals and optima of turnover rates should correspond to the results of other Bayesian calibrations done for similarly structured two-pool models. If such relations could be confirmed, this would point towards fundamental insights about the intrinsic SOM turnover in temperate agroecosystems.

## 2    Material and Methods

### 120    2.1    Study sites and data used for modeling

Datasets originating from bare fallow treatments of four different sites with different experimental durations and measurement frequencies were used in this study. Topsoil (0-20 cm) samples were received from the long-term experiments of (a) the "Ultuna continuous soil organic matter field experiment" (established in 1956, with additional samples from 1979, 1995 and 2005 taken in autumn (Kätterer et al., 2011), four replicates), and (b) the

125 Bad Lauchstädt extreme farmyard manure experiment (established in 1983, with additional samples from 2001, 2004 and 2008 taken in autumn (Blair et al., 2006), two replicates) (https://www.ufz.de/index.php?de=37008, date accessed 10.01.2019). Additional data from two medium-term bare fallow experiments (established in autumn 2009 with data until 2016) from Southwest German regions was included. In these experiments three fields in the region of (c) the Kraichgau and three fields in the region (d) the Swabian Jura, representing different climatic and

130 geological conditions were intensely monitored. The bare fallow plots (5 x 5 m size) in these experiments were established within agricultural fields with three replicates per field (Ali et al., 2015). Up to four topsoil samples (0-30 cm) were taken throughout the year were taken. Further details on all the sites can be found in **Table 1**. All sites had been under cultivation for at least several hundred years prior to establishing the bare fallow plots, which would suggest that steady state could be assumed.

All available bulk soil samples of Ultuna and Bad Lauchstädt were analyzed for total organic carbon and DRIFTS spectra. For Kraichgau and Swabian Jura sites, total organic carbon and DRIFTS spectra were measured about once every two years, while soil microbial biomass carbon (SMB-C) was measured up to four times per year. All bulk soil samples (except for SMB-C) were passed through a 2 mm sieve, then air dried, ball milled (for two minutes) to powder and stored until further analysis. Soil organic carbon (SOC) content was analyzed with a Vario

Max CNS (Elementar Analysensysteme GmbH, Hanau, Germany). Soil samples for DRIFTS analysis were obtained after 24 hr drying at 32, 65 and 105°C. The dried samples were kept in a desiccator until measurement. DRIFTS spectra of bulk soil samples (with four subsamples per sample) were obtained using an HTS-XT microplate extension, mounted to a Tensor-27 spectrometer using the processing software OPUS 7.5 (Bruker Optik GmbH, Ettlingen, Germany). A potassium bromide (KBr) beam splitter with a nitrogen cooled HTS-XT reflection

detector was used to record spectra in the mid infrared range ($4000 - 400$ cm$^{-1}$). Each spectrum was a combination of 16 co-added scans with a 4 cm$^{-1}$ resolution. Spectra were recorded and then converted to absorbance units (AU); the acquisition mode "double-sided, forward-backward" and the apodization function Blackman-Harris-3 were used. After baseline correction and vector normalization of the spectra, areas below absorptions bands of interest were obtained by integration using a local baseline with the integration limits of Demyan et al. (2012). Integrated

band areas of the four subsamples were then averaged. The local baselines were drawn between the intersection of the spectra and a vertical line at the integration limits (3010 – 2800 cm$^{-1}$ for the aliphatic carbon band, 1660 – 1580 cm$^{-1}$ for the aromatic/carboxylate carbon band). Example spectra and integrated band areas are displayed in **Figure S 1**. The integration limits were selected with the goal to reduce signal interference from water and minerals, using spectra of pure substances, clay minerals and from DRIFTS spectra gained during heating samples up to 700°C (Demyan et al., 2013). Particularly, the mineral interference close to the 1620 cm$^{-1}$ band makes accurate selection of integration limits necessary, so that only its top part (assumed to consist mostly of aromatic/carboxylate carbon) is selected. In the case of our samples, the selected specific band area of the 1620 cm$^{-1}$ band accounted for approximately 10 to 30 % of the band area of the larger surrounding band (**Figure S 1**, ca. 1755-1555 cm$^{-1}$). Integration limits were chosen so that the band area best correspond to the portion that is lost with combustion or chemical oxidation (Demyan et al., 2013; Yeasmin et al., 2017). A strong correlation between the DSI and the percentage of centennially persistent SOC (r = 0.84) from the combined long term experiments used in this study (using values of centennially persistent SOC from Cécillon et al., 2018; and Franko and Merbach, 2017), showed that the DSI selected in this manner did in fact explain a large portion of the SOC quality change across sites (**Figure S 2**).

Additionally, soils from the experiments in Kraichgau and Swabian Jura were analyzed for SMB-C using the chloroform fumigation extraction method (Joergensen and Mueller, 1996). Briefly, field moist samples were transported to the lab in a cooler, with extractions beginning within 24 hours after field sampling and the final SMB-C values corrected to an oven-dried (105° C) basis. The SMB-C was measured two to four times throughout the whole year. Stocks of SOC and SMB-C for 0-30 cm were calculated by multiplying the percentage of SOC and SMB-C with the bulk density and sampled layer thickness (**Table 1**), respectively. Bulk density was assumed constant for Bad Lauchstädt, Kraichgau and Swabian Jura, while for Ultuna the initial 1.44 Mg m$^{-3}$ (Kirchmann et al., 2004) in the beginning was used for all but the last measurement, where 1.43 Mg m$^{-3}$ (Kätterer et al., 2011) was used. Due to low coarse fragment contents (< 5 % for Swabian Jura 3, < 2 % for Swabian Jura 1 and < 1 % for the other six sites), and because changes in stone content throughout the simulation periods are unlikely, no correction for coarse fragment content was done.

### 2.2    Description of the simulation model: DAISY Expert-N 5.0

All simulations were conducted using the DAISY SOM model (Hansen et al., 2012) integrated into the Expert-N 5.0 modeling framework. Expert-N 5.0 allows a wide range of soil, plant and water models to be combined and interchanged (Heinlein et al., 2017; Klein et al., 2017; Klein, 2018). Expert-N can be compiled both for Windows and Linux systems. The DAISY model consists of two pools (fast and slow cycling) for each of the measurable fractions of 1) litter, 2) SMB and 3) stabilized SOM (**Figure 1**). Due to bare fallow, litter pools were disregarded in this study and the focus was on initializing the two SOM pools. A detailed description of the DAISY SOM submodule as it was implemented into the Expert-N 5.0 framework can be found in Mueller et al. (1997). The additional modules available for selection in the Expert-N 5.0 framework consist of a selection of established models for all simulated processes in the soil-plant continuum. The evaporation, ground heat, net radiation, and emissivity were simulated according to the Penman-Monteith equation (Monteith, 1976). Water flow through the soil profile was simulated by the Hydrus-flow module (van Genuchten, 1982) with the hydraulic functions

according to Mualem (1976). Heat transfer through the soil profile was simulated with the DAISY heat module (Hansen et al., 1990). In the first step of the DSI evaluation, simulations were conducted with two established parameter sets for DAISY SOM. The first set was from Mueller et al. (1997) and was a modification of the original parameter set of turnover rates reported by Jensen et al. (1997). The second set was established after calibrations made by Bruun et al. (2003) using the Askov Long-Term Experiments, in which they introduced considerable changes to the turnover rates of the slow SOM pool and the humification efficiency. An equation developed by Bruun and Jensen (2002) was used to compute the proportions of the slow and fast cycling SOM pools for both parameter sets at steady state (see next section). Parameters of both sets are given in **Table 2**.

For simulating soil temperature and moisture in Expert-N, daily averages of radiation, temperature, precipitation, relative humidity and wind speed are needed. For the long-term experiments they were extracted from the nearest weather station with complete data (Ultuna source: Swedish Agricultural University (SLU), ECA Station ID #5506, Elevation: 15 m, Lat: 59.8100 N, Long: 17.6500 E; Bad Lauchstädt source: Deutsche Wetter Dienst (DWD) Station #2932, Elevation: 131 m, Lat: 51.4348 N, Long: 12.2396 E, Locality name: Leipzig/Halle). For the fields of the Kraichgau and Swabian Jura, the driving variables were measured by weather stations installed next to eddy covariance stations located at the center of each field. Details on the measurements, instrumentation as well as gap filling methods of those eddy covariance weather stations are described in Wizemann et al. (2015).

### 2.3 SOM pool initializations with the DRIFTS stability index and at steady state

Measured bulk soil SOC includes SMB-C, therefore the amount of SOC in the fast and slow cycling SOM pools combined consists of bulk soil SOC minus measured SMB-C. Partitioning of measured SMB-C into slow (90%) and fast (10%) cycling microbial pools was done similar to Mueller et al. (1998).

The remaining carbon (difference between bulk soil SOC and SMB-C) was divided between fast and slow cycling SOM pools by either the DRIFTS stability index (DSI), or according to the steady state assumption. For steady state division, the equation of Bruun and Jensen (2002) was used, which estimates the fraction of SOM in the slow pool from the model parameters under an assumed steady state:

$$\text{slow SOM fraction } = \frac{1}{1+\frac{k_{SOM\_slow}}{f_{SOM\_slow} * k_{SOM\_fast}}} \tag{1}$$

with $k_{SOM\_slow}$ and $k_{SOM\_fast}$ representing the turnover (per day) of the slow and fast SOM pools respectively, and $f_{SOM\_slow}$ representing the fraction of the fast SOM pool directed towards the slow SOM pool (humification efficiency). This resulted in 83 % of SOM in the slow pool for the original DAISY turnover rates and 49 % in the slow pool for the Bruun et al. (2003) turnover rates (**Table 2**). For the DSI initialization, the ratio of the area below the aliphatic absorption bands to the area below the aromatic/carboxylate absorption band was used as the ratio of SOM in the fast to SOM in the slow cycling SOM pool:

$$\frac{\text{fast SOM}}{\text{slow SOM}} = \frac{\text{A\_2930cm}^{-1}}{\text{A\_1620cm}^{-1}} = DSI \tag{2}$$

Thus, analogous to **Equation 1**, the fraction of SOM in the slow pool was calculated with the formula

$$\text{slow SOM fraction} = \frac{\text{A\_1620cm}^{-1}}{\text{A\_1620cm}^{-1} + \text{A\_2930cm}^{-1}} \tag{3}$$

With A_2930 cm$^{-1}$ and A_1620 cm$^{-1}$ being the specific area under the aliphatic and aromatic/carboxylate band (described in section 2.1). The remaining carbon was allocated to the fast SOM pool. As was mentioned before, three different data inputs for the DSI were used, obtained at drying temperatures of 32, 65 and 105°C, in order to test which drying temperature derived the best proxy for modeling. An example of the change of DRIFTS spectra occurring after several years of bare fallow can be found in **Figure 2**. All DSI model initializations were simulated with both published sets of model parameters. Steady state initializations using **Equation 1** were only simulated with the corresponding parameter set from which they were calculated.

### 2.4    Statistical evaluation of model performance

Statistical analysis was performed with SAS version 9.4 (SAS Institute Inc., Cary, NC, USA). To compare different model initializations, a statistical analysis of squared model errors (SME) was conducted:

$$SME_x = (obs_x - pred_x)^2 \tag{4}$$

with $obs_x$ being the observed value, $pred_x$ the predicted value and $x$ the simulated variable of interest. A linear mixed model with $SME_x$ as response was then used to test for significant differences between initialization methods. This approach allowed us to make use of the statistical power of the three Kraichgau and Swabian Jura fields to analyze which initialization was most accurate and to evaluate the trend of the model error with increasing simulation time. In some cases, $SME_x$ was transformed to ensure a normal distribution of residuals (square root transformation for Ultuna SOC and Kraichgau/Swabian Jura SMB-C and fourth root for Kraichgau/Swabian Jura SOC), which was checked by a visual inspection of the normal QQ plots and histograms of residuals (Kozak and Piepho, 2018). Random effects were included to account for temporal autocorrelation of $SME_x$ within (a) the same field and (b) the same simulation. The model reads as follows:

$$y_{ijkl} = \phi_0 + \alpha_{0i} + ß_{0j} + \gamma_{0ij} + \phi_1 t_k + \alpha_{1i} t_k + ß_{1j} t_k + \gamma_{1ij} t_k + u_{kl} + u_{ijkl} \tag{5}$$

where $y_{ijkl}$ is the $SME_x$ of the simulation using the $i$th initialization with the $j$th parameter set, at the $k$th time on the $l$th field, $\phi_0$ is an overall intercept, $\alpha_{0i}$ is the main effect of the $i$th initialization, $ß_{0j}$ is the main effect $j$th parameter set, $\gamma_{0ij}$ is the $ij$th interaction effect of initialization x parameter set, $\phi_1$ is the slope of the time variable $t_k$, $\alpha_{1i} t_k$ is the interaction of the $i$th initialization with time, $ß_{1j} t_k$ is the interaction of the $j$th parameter set with time, $\gamma_{1ij} t_k$ is the $ij$th interaction effect of initialization x parameter set x time, $u_{kl}$ is the autocorrelated random deviation on the $k$th time in the $l$th field and $u_{ijkl}$ is the autocorrelated residual error term corresponding to $y_{ijkl}$. The detailed SAS code can be found in the supplementary material. For Ultuna and Bad Lauchstädt, the $u_{kl}$ term was left out, as both trials only had one field. As the Kraichgau and Swabian Jura had the exact same experimental setup and duration, these sites were jointly analyzed in the statistic model, but due to completely different setups and durations, this was not possible for Bad Lauchstädt and Ultuna. The full models with all fixed effects were used to compare different correlation structures for the random effects including (i) temporal autocorrelation (exponential, spherical, Gaussian), (ii) compound symmetry, (iii) a simple random effect for each different field and simulation, (iv) a random intercept and slope of the time variable (with allowed covariance between both) for each field and initialization method. A residual maximum likelihood estimation of model parameters was used and the best fitting random effect structure for this model was selected using the Akaike Information Criterion as specified by Piepho et al. (2004). Then a stepwise model reduction was conducted until only the significant effects

($p < 0.05$) remained in the final statistical model. Because a mixed model was used, the Kenward-Roger method was applied for estimating the degrees of freedom (Piepho et al., 2004) and to compute post hoc Tukey-Kramer pairwise comparisons of means.

## 2.5 Model optimization and observation weighting for Bayesian calibration

Optimization of parameters $k_{SOM\_slow}$, $k_{SOM\_fast}$ and the humification efficiency ($f_{SOM\_slow}$) was performed using a Bayesian calibration approach. These parameters were chosen as only they have a considerable impact on the rate of native SOM loss (see further details in the supplementary chapter S 12.2 ). The Bayesian calibration method uses an iterative process to simulate what the distribution of parameters would be, given the data and the model. It combines a random walk through the parameter space with a probabilistic approach on parameter selection.

The Differential Evolution Adaptive Metropolis algorithm (Vrugt, 2016) implemented in UCODE_2014 (Lu et al., 2014; Poeter et al., 2014) was used for the Bayesian calibration in this study. As no Bayesian calibration of DAISY SOM parameters has been done before, noninformative priors were used. The main drawback of noninformative priors is that they can have longer computing times, but as was shown by Lu et al. (2012) with sufficient data and simulation durations, the posterior distributions are very similar to using informed priors. Ranges were set far beyond published parameters with $1.4 * 10^{-2}$ to $1.4 * 10^{-6}$ d$^{-1}$ for $k_{SOM\_fast}$ and $1.4 * 10^{-3}$ to $5 * 10^{-7}$ d$^{-1}$ for $k_{SOM\_slow}$. The parameter $f_{SOM\_slow}$ had to be more strongly constrained as without constraints it tended to run into unreasonable values up to 99 % humification. The limits were therefore set to 0.05 to 0.35, which is +/- 5 % of the two published parameter sets and represents the upper boundaries of other similar models (e.g. Ahrens et al., 2014). The default UCODE_2014 Gelman-Rubin criterion (Gelman and Rubin, 1992) value of 1.2 was chosen for the convergence criteria. A total of 15 chains were run in parallel with a timestep of 0.09 days in Expert-N 5.0 (this was the largest timestep and fastest computation, where the simulation results of water flow, temperature and hence SOM pools was unaltered compared to smaller timesteps). It was ensured that at least 300 runs per chain were done after the convergence criterion was satisfied.

In Bayesian calibration, a proper weighing of observations is needed in order to achieve a diagonal weight matrix of residuals (proportional to the inverse of the variance covariance matrix), and to ensure that residuals are in the same units (Poeter et al., 2005, p18 ff). This included several steps. A differencing removed autocorrelation in the individual errors in each model run of the Bayesian calibration itself (the first measurement of each kind of data at each field was taken as raw data, for any repeated measurement the difference from this first measurement was taken instead of the raw data). Details on differencing are provided in chapter 3 of the UCODE_2005 manual (Poeter et al., 2005). To account for varying levels of heterogeneity of different fields in the weighting, a mixed linear model was used to separate the variance of observations from different fields originating from natural field heterogeneity from the variance originating from measurement error. To do so, a linear mixed model with random slope and intercept of the time effect for each experimental plot was fitted to the SOC, SMB-C and DSI data for each field individually:

$$y_{kl} = \phi_0 + \phi_1 t_k + u_l + u_k + u_{kl} \tag{6}$$

where $y_{kl}$ is the modeled variable at the $k$th time on the $l$th plot, $\phi_0$ is the intercept, $\phi_1$ is the slope of the time variable $t_k$, $u_l$ is the random intercept, $u_k$ is the autocorrelated random deviation of the slope and $u_{kl}$ is the autocorrelated residual error term corresponding to $y_{kl}$.

The error variance of each type of measurement (DSI, SMC-C, SOC) at each field $\sigma_{fM}{}^2 = \sigma_{u_k}^2 + \sigma_{u_{kl}}^2$ was then used for weighting of observations, excluding the field variance $\sigma_{u_l}^2$ from the weighting scheme. This error variance was used in UCODE_2014 to compute weighted model residuals for each observation as follows:

$$w\_SME_x = \frac{(obs_x - pred_x)^2}{\sigma^2{}_{fM}} \tag{7}$$

where $w\_SME_x$ is the weighted squared model residual, $obs_x$ is the observed value, $pred_x$ is the predicted value and $\sigma^2{}_{fM}$ is the error variance of the $M$th type of measurement at each field. All $w\_SME_x$ are combined to the sum of squared weighted residuals, which is the objective function used in UCODE_2014 (Poeter et al., 2014). By this procedure, observations with higher measurement errors have a lower influence in the Bayesian calibration.

Since the medium-term experiments had a much higher measurement frequency, it was also tested if giving each experiment the same weight would improve the results of the Bayesian calibration (equal weight calibration). In this case an additional group weighting term was introduced for groups of observations, representing different datasets at the different sites. This weighting term is internally multiplied with each $w\_SME_x$ in UCODE_2014 and was calculated as

$$w\_G_x = \frac{1}{(n_{obs} * n_{par} * n_f)} \tag{8}$$

where $w\_G_x$ is the weight multiplier for each observation, $n_{obs}$ is the number of observations per parameter, $n_{par}$ is the number of parameters per field, and $n_f$ is the number of fields per site. This weighing assures that with the exact same percentage of errors, each site would have the exact weight of 1.

The influence of several factors was assessed in this Bayesian calibration: the use of individual sites compared to combining sites, including an equal weight (EW, as described above) vs original weight (OW) weighting only by error variance, and the effect of in/excluding the DSI (+/- DSI) in the Bayesian calibration. Therefore, seven Bayesian calibrations were conducted in total: four for each individual site with original weight and DSI, i.e., 1) Ultuna, 2) Bad Lauchstädt, 3) Kraichgau, 4) and Swabian Jura, 5) equal weight calibration for all sites combined using DSI, 6) original weight calibration for all sites combined without using DSI in the Bayesian calibration (only for initial pool partitioning) and 7) original weight calibration for all sites combined using the DSI. The comparison of these seven Bayesian calibrations was designed to assess the effect of the site on the calibration, as well as the effect of the DSI and of user weighting decisions.

## 3    Results

### 3.1    Dynamics of SOC, SMB-C and DRIFTS during bare fallows

All bare fallow plots lost SOC over time with the severity of SOC loss varying between soils and climates at the different sites. The Bad Lauchstädt site experienced the slowest carbon loss (7% of initial SOC in 26 years), while

SOC at Ultuna and Kraichgau was lost at much faster rates (Ultuna - 39% of initial SOC in 50 years, Kraichgau on average 9% of initial SOC in 7 years) (**Table 3**). In the Swabian Jura field 1 the SOC loss was comparable to that of Kraichgau (about 10% of initial SOC in 7 years), but was much less in fields 2 and 3. Some

330 miscommunications with the field owner's contractors led to unwanted manure addition and fields ploughing in Swabian Jura field 2 and 3 in 2013, hence results of these two fields after the incident in 2013 were excluded. The DRIFTS spectra revealed that the aliphatic carbon band area (2930 cm$^{-1}$) decreased rather fast after the establishment of bare fallow plots while the aromatic/carboxylate band area (1620 cm$^{-1}$) showed only minor changes and no consistent trend (**Figure 2**). The assumed fraction of SOC in the slow SOM pool according to the

335 DSI at 105°C changed from the initial range of 54 to 80 % to the range of 76 to 99% at the end of the observational period (**Table 3**, **Figure S 3**). The SMB-C reacted even more rapidly to the establishment of fallows and halved on average for all fields within 7 years duration (**Table 3**).

### 3.2 Comparison of the different model initializations

The observed trend of SOC loss with ongoing bare fallow duration was also found in all simulations (**Figure 3**

**and Figure S 4**). For Ultuna, simulated SOC loss in all cases underestimated measured loss, while for Bad Lauchstädt, simulated SOC losses consistently overestimated measured losses. At Kraichgau sites SOC loss was underestimated by the models, but with the Bruun (2003) parameter set yielding simulated values closer to actual measurements. In the Swabian Jura, both parameter sets underestimated SOC loss. The decline of SMB-C in the Kraichgau and Swabian Jura (**Figure 4**) occurred more rapidly than that of SOC, though SMB-C had higher

variability of measurements. The parameter sets with steady state assumptions marked the upper and lower boundaries of the SMB-C simulations but the DRIFTS stability index (DSI) initializations were closer to the measured values (with exception of Swabian Jura field 3). For brevity only simulations of field 1 for Kraichgau and Swabian Jura are shown. Simulation results for fields 2 and 3 are found in the supplemental material (**Figure S 5** for SOC simulations and **Figure S 6** for SMB-C).

The statistical analysis of the model error revealed a site dependency of the effect of the parameter set. The three-way interaction of initialization, parameter set and time $\gamma_{1ij}t_k$ was significant for all but Bad Lauchstädt SOC, where only the parameter set had a significant effect. In the case of Bad Lauchstädt, the model error was significantly lower with the slower Mueller (1997) SOM turnover parameter set, while for the rest of tested cases, the faster Bruun (2003) set performed significantly better (**Table 4**). For Ultuna and Kraichgau + Swabian Jura

SOC, the steady state assumption with Mueller (1997) parameters had the highest model error, while the steady state assumption with Bruun (2003) parameters had the lowest model error of all simulations, being similar to DSI initializations at Kraichgau and Swabian Jura. However, there was a statistical significantly lower SOC model error with DSI using 105°C drying temperature than the lower drying temperatures for the Ultuna site. For SMB-C simulations at the Kraichgau + Swabian Jura sites, however, the errors were lowest for the DSI initialization using

the 105° C drying temperature with Bruun (2003) parameters and significantly lower than both steady state initializations. Of the DSI initializations using different drying temperatures, the model error was always lowest when using the 105°C drying temperature initialization compared to 32°C and 65°C (significant for Ultuna, as well as for Kraichgau + Swabian Jura SMB-C using Mueller (1997) parameters). As initializations with DSI using

105°C drying temperature consistently performed best of all three DSI initializations, only DSI spectra of soils dried at 105°C were used for the Bayesian calibration.

### 3.3    Informed turnover rates of the Bayesian calibration

The posterior distribution of parameters from the Bayesian calibration differed considerably between the different calibrations for individual sites, but there were also differences between different weighting schemes or when performing the Bayesian calibration without using the DSI (**Figure 5**). The highest probability turnover of the fast SOM pool ($k_{SOM\_fast}$) was 1.5 and 3 times faster for Ultuna and Kraichgau, respectively, when compared to initial rates ($1.4 * 10^{-4}$ $d^{-1}$ for both parameters sets), which fitted well for Bad Lauchstädt and Swabian Jura. For the slow SOM pools ($k_{SOM\_slow}$) the Bad Lauchstädt, Kraichgau and Swabian Jura site calibrations were in between the two published parameter sets, but tended towards the slower rates ($2.7 * 10^{-6}$ $d^{-1}$ by Mueller (1997)), while the optimum for Ultuna was exactly at the fast rates of Bruun (2003) ($4.3 * 10^{-5}$ $d^{-1}$). The humification efficiency ($f_{SOM\_slow}$) was not strongly constrained in the Bayesian calibration, except for the Kraichgau site, where it ran into the upper boundary of 0.35. This trend towards higher humification existed also for the other sites, but to a lesser extent than for Kraichgau.

The different calibrations of the combination of all sites under different weightings and with or without the DSI led to considerable differences in the posteriors (**Figure 5**). When combining the sites with the artificial equal weighting, the posterior distribution of all three parameters was the widest, basically covering the range of all four site calibrations. With the original weighting scheme, only informed by the variance of the data, the posteriors were narrower for all parameters, with the optima of $k_{SOM\_fast}$ being slightly faster than the two (similar) published rates. The optima of $k_{SOM\_slow}$ were slightly slower than Bruun (2003) but much faster than Mueller (1997) and $f_{SOM\_slow}$ was even above the higher Bruun (2003) value of 0.3. The use of the original weighting scheme without the use of the DSI in the Bayesian calibration did not constrain the $f_{SOM\_slow}$ at all and had faster $k_{SOM\_slow}$ and slower $k_{SOM\_fast}$ than the one using the DSI. Both these Bayesian calibrations using the original weighting (with and without DSI) showed a trend towards slightly faster turnover than suggested by Bruun (2003).

There was a strong negative correlation between $k_{SOM\_fast}$ and $k_{SOM\_slow}$ parameters for all but the Bad Lauchstädt calibration (**Figure S 7**). When DSI was not included in the Bayesian calibration, this negative correlation was stronger than when it was included (**Figure 6**). The parameters $k_{SOM\_fast}$ and $f_{SOM\_slow}$ were always positively correlated, most strongly for Kraichgau (0.49) and Swabian Jura (0.38), but only weakly for the long-term sites. The correlations between the parameters $k_{SOM\_slow}$ and $f_{SOM\_slow}$ were generally low and both positive and negative. The parameters with the highest probability density of the calibrations combining all sites for $f_{SOM\_slow}$, $k_{SOM\_fast}$ and $k_{SOM\_slow}$ in that order were 0.34, $2.29 * 10^{-4}$, $3.25 * 10^{-5}$ for the original weight calibration and 0.06, $9.58 * 10^{-5}$ and $5.54 * 10^{-5}$ for the calibration using original weights and no DSI. These results suggest that turnover rates of $k_{SOM\_slow}$ could be similar or faster than $k_{SOM\_fast}$ without the use of the DSI. About 10 % of simulations of the Bayesian calibration without DSI had even a faster $k_{SOM\_slow}$ than $k_{SOM\_fast}$.

### 4    Discussion

### 4.1 How useful is the DRIFTS stability index?

A search for suitable proxies for SOM pool partitioning into SOM model pools that correspond to measurable and physicochemical meaningful quantities is of high interest (Abramoff et al., 2018; Bailey et al., 2018; Segoli et al., 2013). The results of this study confirm the hypothesized usefulness of the DSI proxy assessing the current state of SOM for pool partitioning to model SOC for several soils across Europe. This is particularly relevant, given that changes in crop genotype and rotation, agricultural management, and the rise of average temperatures in recent decades as well as land use changes, such as draining of soils or deforestation, in recent centuries have altered the quality and quantity of carbon inputs to soil. Consequently, the steady state assumption for model initialization is not likely to be valid. Demyan et al. (2012) showed that with a careful selection of integration limits for absorbance band areas, the DSI through identifying organic contributions in DRIFTS spectra is a sensitive indicator of SOM stability if mineralogy is similar (despite acknowledged mineral interference). Combined with a higher temperature (105 °C) for soil drying prior to DRIFTS analysis, a strong correlation between the portion of centennially persistent SOC and the DSI (**Figure S 2**) was found in our study which supports the hypothesis that DSI might be of general applicability across sites. Results from modeling corroborated the usefulness of the DSI for SOM pool partitioning for soils of different properties across Europe. The statistical analysis of the model error for both SOC and SMB-C showed clearly that the DSI can improve poor model performance, especially when the slower turnover rates of Mueller (1997) were used. When model performance is already satisfactory, the natural variability of the DSI can make model performance worse, as in the case of Ultuna SOC with Bruun (2003) parameters, but this reduction was minor compared to the improvement the DSI had over steady state assumptions at Ultuna with Mueller (1997) rates. The better results for Ultuna with the Bruun (2003) steady state might also just be an effect of turnover times still being too slow and hence the more SOC in the fast pool, the faster turnover is in general and the lower the model error. This was also indicated by faster optima by the Bayesian calibration compared to both published turnover rates. In the case of the Chernozem of Bad Lauchstädt, only turnover rates had an influence on model performance and its SOC turnover was overestimated by both parameter sets (**Figure 3**). It was previously suggested that the high SOC storage capacity of this site is a result of cation-bridging due to a high content of adsorbed cations (Ellerbrock and Gerke, 2018). Additionally, there is evidence for black carbon at the site (e.g. the high thermal stability found by Demyan et al., 2013). Therefore, a possible reason for an overestimation of SOC turnover in Bad Lauchstädt might be, that DAISY only considers clay content as a stabilizing mechanism. Nevertheless, the use of DSI also was suitable for Bad Lauchstädt, as there was no significant difference in model performance compared to steady state.

The range of different sites, soils, and climatic conditions of Europe represented within this study suggests the robustness of the DSI as a proxy for SOM quality and SOM pool division for a large environmental gradient. Hence, it would be an improvement over assuming steady state of SOM wherever there is a lack of detailed information of carbon inputs and climatic conditions. Considering the timescales at which SOM develops, this is almost anywhere, as detailed data is available at best for <200 years, which is not even one half-life of the slow SOM pool.

So far, studies that assessed SOM quality and pool division proxies, either using thermal stability of SOM (Cécillon et al., 2018) or size-density fractionation (Zimmermann et al., 2007), only indirectly related the proxies to inversely

modeled SOM pool distributions, using machine learning and rank correlations. In contrast, our study showed that the DSI is a proxy which can be directly used for pool initialization. The DSI also makes sense from the perspective of energy content, as microorganisms can obtain more energy from the breakdown of aliphatic than aromatic/carboxylate carbon compounds (e.g. Good and Smith, 1969), and therefore aliphatic carbon is primarily targeted by microorganisms (hence have faster turnover) as previously shown for bare fallows (Barré et al., 2016).

The two distinct absorption bands for aliphatic and aromatic/carboxylate carbon bonds of the DSI fit well to the two SOM pool structure of DAISY and the simulation of carbon flow through the soil in DAISY is very similar to several established SOM models such as SoilN, ICBM and CENTURY. It is therefore likely that with calibration, the DSI could be used as a general proxy for SOM models with two SOM pools and a humification efficiency ($f_{SOM\_slow}$ in DAISY). The parameter correlations between $k_{SOM\_slow}$, $k_{SOM\_fast}$ and $f_{SOM\_slow}$ according to the Bayesian calibrations also suggest that without a pool partitioning proxy, modifying any one parameter can lead to similar results in terms of SOC and SMB-C simulation. A clear distinction between fast and slow pools needs a pool partitioning proxy as can be seen by faster $k_{SOM\_slow}$ than $k_{SOM\_fast}$ for some of the simulations of the Bayesian calibration without using DSI. Assigning the DSI to DAISY reduced parameter correlations and led to a clear distinction between fast and slow SOM pools.

The DRIFTS absorption band for aliphatic carbon is most resolved when applying a 105°C drying temperature to samples prior to analysis (Laub et al., 2019). The current study's modeling results corroborated the finding that the DSI should be obtained from measurements after drying at 105 °C with the performance of the DRIFTS initializations being always in the order 105°C > 65°C > 32°C drying temperature (differences being sometimes but not always significant).

Compared with the other proxies for SOM quality discussed above, the measurements by DRIFTS are inexpensive, relatively simple, and the equipment of the same manufacturer is standardized. This should also constrain variability between different laboratories and be attractive for large-scale applications with large sample numbers, for example to initialize simulations at the regional scale. However, for standardization of the DSI for model initialization one needs to address how the type of spectrometer (e.g. detector type) influences the spectra, if water and mineral interferences (Nguyen et al., 1991) in the spectra can be further reduced and if a mathematical standardization of the spectra and DSI (across instruments and water contents) is possible. While a complete elimination of mineral interference is not possible, a careful selection of integration limits and the use of a local baseline minimizes mineral interference of DRIFS spectra from bulk soils. This mostly selects the top part of the 1620 cm$^{-1}$ band area, which corresponds to the part that is reduced or completely lost when SOC is destroyed (Demyan et al., 2013; Yeasmin et al., 2017). Other approaches such as spectral subtraction of ashed samples or HF destruction of minerals prior DRIFTS analysis have been developed in the attempt to obtain spectra of pure SOC. All are rather labor intensive and still produce artifacts, as it is not possible to destroy only the minerals or only the SOC without altering the respective other fraction (Yeasmin et al., 2017). Hence, we think that the selected integration limits might represent at this point the most feasible option for obtaining a robust and cost-effective proxy of SOC quality for modeling. The strong correlation of DSI and centennially persistent SOC as well as the model results of this study seem to corroborate this. The method of DSI estimation might be improved by a study of the best integration limits optimizing the fit of the DSI and centennially persistent SOC, which would require

more bare fallow experiments than in this study. From a conceptual perspective DSI probably relates mainly to chemical recalcitrance of SOM present in different SOM fractions. In that respect it is different from physical light/heavy fraction separation approaches as each of these fractions is very heterogeneous. For example, the light fraction has strong absorbance at both aliphatic and aromatic/carboxylate carbon bands (Calderón et al., 2011), so it could be that within each fraction, aliphatic carbon is preferentially consumed by microorganisms. Thus, DSI reflects physicochemically stabilized SOC (mainly mineral-association in the case of bare soils) as also suggested by the correlation of the ratio of 1620 cm$^{-1}$/2930 cm$^{-1}$ absorption bands to the ratio of mineral associated carbon/light fraction carbon (Demyan et al. 2012). The relationship to mineral-association in many models is represented by a texture adjustment factor. On the other hand, DSI does not directly relate to aggregated (i.e. occluded) SOM, and its applicability in models focusing on aggregation needs to be evaluated (i.e. by a separate spectral analysis of occluded and remaining fractions.

The recent coupling of pyrolysis with DRIFTS (Nkwain et al., 2018) might be a further analytical advancement of the DSI, as it overcomes mineral interferences in the spectra. However, this technique is more complex due to a larger number of visible organic absorption bands, including $CO_2$ that develops from the pyrolysis, which makes it not easily applicable to established two-pool models such as DAISY. In addition, a considerable portion (30 – 40 %) of SOM is not pyrolyzed and therefore not recorded in the spectra. In summary, despite of the acknowledged shortcomings, the DSI was useful to partition SOM between pools, even more so, when the optimized parameters for the DSI will be used for future applications. It seems more robust than steady state or long-term spin-up runs which rely on strong assumptions. Further tests are needed before using the DSI for mineralogy that differs considerably from the soils of this study.

### 4.2    Parameter uncertainty as estimated with Bayesian calibration

According to our Bayesian calibrations, a wide range of parameter values are possible for DAISY going far beyond the initial published parameter sets. By combining various sites and including meaningful proxies, such as the DSI, the parameter uncertainty and equifinality could be reduced and the credibility intervals narrowed. The predictions of mechanistic models usually fail to account for the three main statistical uncertainties of (1) inputs, (2) scientific judgments resulting in different model setups and (3) driving data (Wattenbach et al., 2006). However, with a Bayesian calibration framework such as implemented in UCODE 2014, almost any model can be made probabilistic, so uncertainties of parameters and outputs can be assessed, even for projections into the future (Clifford et al., 2014). As this study focused on Bayesian calibration and we used an established model, we mainly address parameter uncertainty, although input uncertainty was also included through the weighting process. We clearly demonstrated an effect of the individual site used for Bayesian calibration on the resulting model parameters and uncertainties. Similarly diverging site specific turnover rates were also found by Ahrens et al. (2014) in a study of soil carbon in forests. Diverging results for different sites generally point towards a need for a better understanding of the modeled system and model improvements (Poeter et al., 2005), but this often requires a deeper understanding of the system and new measurements – hence it is not always feasible. A Bayesian calibration asks the question: "What would be the probability distribution of parameters, given that the measured data should be represented by the selected model?". Hence, if only one site is used, it can only answer this question for that specific site. As this study showed, the parameter set could then be highly biased for other sites. For a

more robust calibration, several sites should be combined to obtain posterior distributions of parameters for a gradient of sites, though this might reduce model performance for individual sites. The introduction of the equal weighting scheme, which gave similar weights to the different sites, highlights how much bias may be introduced by user decisions of artificial weighting: this Bayesian calibration parameter set had the highest uncertainties and it appears as if the Ultuna site had by far the strongest influence. In contrast to that, the combination of all four sites with the original weights based on the error variances or measurements led to a very clear reduction of parameter uncertainty and the narrowest parameter credibility intervals (**Figure 6 a** compared to **b** and **c**).

The results of the statistical analysis of model errors (**Table 4**) suggests that the DSI is suitable for SOC model pool initialization. This was corroborated by the Bayesian calibration, as the inclusion of the DSI narrowed credibility intervals for the slow SOM pool turnover and humification efficiency and reduced the correlation between fast and slow SOM turnover compared to the simulation without the DSI as constraint. Especially in the case of the clear differentiation between $k_{SOM\_slow}$ and $k_{SOM\_fast}$, our results show the advantage of attaching a physiochemical meaning to the pools that was not provided before. Other effective approaches, such as time series of $^{14}C$ data could be combined with the DSI for better results.

Of all three parameters, the humification efficiency ($f_{SOM\_slow}$) was the only parameter that consistently ran into the upper boundaries, set to 35 %. In fact, initial calibrations were done where $f_{SOM\_slow}$ was constrained to 95 %; even then, it tended to run into that constraint (**Figure S 8**) and led to much faster turnover rates ($k_{SOM\_slow}$) than were published before. These values of $f_{SOM\_slow}$ were much greater than the 10 % for the Mueller (1997) dataset, 30 % for Bruun (2003), and other published two pool models. Therefore, we considered the cause of the poorly constrained $f_{SOM\_slow}$ parameter to be a model formulation problem, which did not depend on whether the DSI was included in the Bayesian calibration or not. Only when the humification efficiency was restricted in the Bayesian calibration, the turnover of fast and slow SOM aligned with the earlier published rates. If a parameter is problematic, such as $f_{SOM\_slow}$ it could mean that there is a lack of data. However, if parameters are constrained, but run into implausible values, it usually means that the model structure is suboptimal (Poeter et al., 2005) and should be altered.

### 4.3 Model structure determines SOM turnover times in two-pool models

The rate of SOM decomposition remains of major interest, especially with respect to the potential of SOM as a global carbon sink (Minasny et al., 2017). Some of the first conceptual approaches proposed SOM pools with residence times of 1000 years and longer (e.g. in CENTURY, Parton et al., 1987), but the SOM models were calibrated to fit data measured in long-term experiments that included vegetation. The pool structure of early SOM models such as DAISY and CENTURY were rather similar as were the turnover rates of SOM pools (see summary in **Table 5**). An improved understanding of actual amounts of carbon inputs to the soil, which remain challenging to measure, led to faster turnover rates in more recent model versions (e.g. by Bruun, 2003). The reason is probably that inputs of carbon and nitrogen to the soil were initially underestimated as it is very difficult to measure root turnover and rhizosphere exudation inputs without expensive in situ $^{13}C$ or $^{14}C$ labeling. The underestimated inputs were then likely counterbalanced in the model calibration by slower turnover rates resulting in acceptable model outputs (SOM dynamics and $CO_2$ emissions) for the time being. However, as our summary of more recent studies underlines (**Table 5**), the earlier published turnover rates seem to be subject to a systematic underestimation. As

the comparison of our Bayesian calibration to other recent Bayesian calibration studies suggests, the relatively fast turnover rates of this study are in alignment with other recent findings (**Table 5**), as all five examples have published turnover rates for the slow SOM pool, which are at least one order of magnitude faster than early assumptions from the 1980s and 90s.

It is critical to understand model uncertainties and to test fundamental assumptions of how SOM is transferred between the pools (Sulman et al., 2018). The comparison between constrained and unconstrained humification efficiency in the Bayesian calibrations suggests that the sequential flow of carbon through the system might be assuming a condensation of stabile carbon that does not actually explain the vast majority of more stable SOM formation. From a theoretical perspective, one may wonder how large amounts of less complex SOM should

become complex SOM without any involvement of living soil organisms. The way that the formation of complex carbon is represented in DAISY is probably a remainder of earlier humification theories from the 1990s that mostly ignored microbe involvement, while most of the recent studies suggest that the vast majority of SOM is of microbial origin (Cotrufo et al., 2013). A simple adaption for two-pool SOM models such as DAISY that include SMB pools could acknowledge this paradigm shift: The partitioning between slow and fast turnover SOM could

be at the death of the microbial biomass (**Figure 7**) without any transfer of SOM from fast to slow pools (a brief test of this new structure is provided in the supplementary material **Figure S 10**). This would also be in alignment with the DSI concept, as aliphatic carbon should not spontaneously transform to aromatic/carboxylate carbon on its own. Then DAISY would fit better to the DSI and other proxies linking measurable fractions to SOM pools (the same is true for CENTURY and other models, which apply the same humification principle). The way that

pools are linked in the current model configuration, is such that the actual turnover time of recalcitrant SOM consists of the turnover of the fast and slow SOM pools combined as it moves through these pools sequentially (**Figure 1**).

How strongly the basic model assumptions influence SOM simulations is also reflected when differences between one- and two-SOM pool models are compared. The turnover rates of the one-pool models are in between those of

575 slow and fast SOM pools. However, our comparison shows that models with similar structure come to similar conclusions for SOM turnover. For example, the one-pool model in Clifford et al. (2014) was quite similar in turnover rates to that in Luo et al. (2016), but does not match well with two-pool models. Then again, the rates for the two-pool models of this study, and the studies by Ahrens et al. (2014) and Hararuk et al. (2017) were very similar in their minima and maxima, for both the slow and fast SOM pools, which shows that only models with a

580 similar number of pools and transformations could be compared.

The 95 % credibility intervals of half-lives in DAISY were in the range from 278 to 1095 years for the slow SOM pool and from 47 to 90 years for the fast SOM pool for the combination of sites presented in this study. If these values were reasonable – and as the three recently published Bayesian calibrations including this study are quite close in turnover rates (**Table 5**), this seems to be the case, SOM could be lost at much faster rates under

585 mismanagement and global warming than earlier modeling results suggest. The rates may also be biased towards an underestimation of turnover, as even with intense efforts it is next to impossible to keep bare fallow plots completely free of vegetation (weeds) and roots from neighboring plots. Recent studies are in alignment with the possibility of relatively fast SOC loss across various scales from field scale (Poyda et al., 2019) to country scale.

For example in Germany, agricultural soils are much more often a carbon source than a sink (Jacobs et al., 2018). This highlights the importance of adequate SOM management and a deeper understanding of the processes at different scales. Especially in the context of understanding the response of SOM to climate change it is not enough if the SOM balance is simulated appropriately, but also fluxes within the plant-soil system need to be quantified. The reason is that under a warmer climate and changing soil moisture levels, the plant-derived carbon inputs will change. Furthermore, soil enzymatic analysis at regional and field level (Ali et al., 2015, 2018) suggest that pools of different complexity have different temperature sensitivities (Lefèvre et al., 2014), which is also realized in new models (Hararuk et al., 2017). If different pools have different responses to temperature, the formula by Bruun and Jensen (2002) for SOM pool distribution could not be used anymore, as it implicitly assumes a similar temperature sensitivity for all pools. In light of this, new proxies such as the DSI, soil fractionation or $^{14}$C use (Menichetti et al., 2016), which could also be combined, are crucial for making SOM pools chemically or physically meaningful and to reduce model uncertainty and equifinality. As the DSI also had a good correlation with structurally protected SOM (Demyan et al., 2012) it could also fit very well to models that directly simulate the protection of SOM as a function of microbial activity (Sulman et al., 2014). A better understanding and the use of meaningful proxies such as DRIFTS, pyrolysis with DRIFTS (Nkwain et al., 2018) or thermal deconvolution (Cécillon et al., 2018; Demyan et al., 2013) in combination with Bayesian calibration and a wide range of long-term experiments are needed. The discrepancy between simulating SOM of tropical and temperate soils, which points towards a lack of understanding of fundamental difference in processes at work on the global scale would be the best test for future proxies and SOM models, which should be facilitated by freely available datasets for model testing and calibration.

## 5    Conclusion

We tested the use of the DRIFTS stability index as a proxy for initializing the two SOM pools in the DAISY model and used a Bayesian calibration to implement this proxy. A statistical analysis of model errors suggested that the use of DRIFTS stability index to initialize the fast and slow SOM pools significantly reduced model errors in most cases, especially those with initially poor performance. DSI therefore seems to be a robust proxy to distinguish between fast and slow cycling SOM in order to initialize two-pool models and adds physicochemical meaning to the pools. As other studies have also shown, statistically sound approaches such as Bayesian calibration are needed to grasp the high uncertainty of SOM turnover, which is often neglected in modeling exercises. The results of the Bayesian optimization procedure further suggest that model performance could be improved by adjusting model parameters (turnover rates, humification efficiency) to the DSI initialization approach. Meaningful proxies such as DRIFTS, physical/chemical fractionation, or $^{14}$C age assessments are likely to be the most robust way to initialize SOM pools but their measurement method needs to be optimized to overcome known constraints, such as water and mineral interference in the case of DSI. The results of this study suggest that the turnover of SOM could be much faster than assumed by commonly used SOM models. For example, the DAISY slow SOM pool half-life estimated in our study ranged from 278 to 1095 years (95 % credibility intervals). The variability of parameters highlights the importance to include meaningful proxies into SOM models and to conduct research on a larger gradient of soils with bare fallow and planted sites, and over longer time frames.

## 6    Acknowledgements

This research was supported by the German Research Foundation (DFG) under the projects PAK 346 and the following FOR1695 "Agricultural Landscapes under Global Climate Change – Processes and Feedbacks on a Regional Scale" within subproject P3 (CA 598/6-1). We would like to thank Elke Schulz from the Department of Soil Ecology, Helmholtz Centre for Environmental Research in Halle/Saale for the provision of samples from Bad Lauchstädt. We would also like to thank Steffen Mehl, from the UCODE development team, for his help with the weighing of observations and the troubleshooting during the setup of UCODE_2014 on the bWUniCluster. Finally, we thank the editor and all reviewers, especially Lauric Cécillon for the fruitful discussions during the review process. The authors acknowledge support by the state of Baden-Württemberg through bwHPC.

## 7    Data availability

Data of SOC from Ultuna and Bad Lauchstädt have already been published in the last decades and are cited in the text. The data of Kraichgau and Swabian Jura has not been published yet, but is provided in the graphs. The raw data which was used in this study is available in the supplement of this article.

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

850

## 9    Tables

**Table 1 Locations, soil type according to (IUSS Working Group WRB, 2007), initial soil organic carbon (SOC) stocks and other properties of the simulated bare fallow study sites**

| Study Site | UTM Degrees Latitude | UTM Degrees Longitude | Soil type | Depth of sampling (cm) | Rep. | Clay (%) | Silt (%) | Initial SOC (%) | Bulk density (Mg/m³) | Initial SOC stocks in the sampled depth at fallow start (Mg/ha) | Year of experiment and bare fallow establishment | Bulk soil samples available from years | Types of available measurements |
|---|---|---|---|---|---|---|---|---|---|---|---|---|---|
| Ultuna[1] | 59.821879 | 17.656348 | Eutric Cambisol | 0 - 20 | 4 | 37 | 41 | 1.50 | 1.44 | 43.22 | 1956 | 1956, 79, 95, 2005 | SOC, DRIFTS |
| Bad Lauchstädt[2] | 51.391605 | 11.877028 | Haplic Chernozem | 0 - 20 | 2 | 21 | 68 | 1.82 | 1.24 | 45.08 | 1985 | 1985, 2001, 04, 08 | SOC, DRIFTS |
| Kraichgau 1 | 48.928517 | 8.702794 | Stagnic Luvisol | 0 - 30 | 3 | 18 | 97 | 0.90 | 1.37 | 37.10 | 2009 | 2009 - 16 | SOC, DRIFTS, SMB-C |
| Kraichgau 2 | 48.927748 | 8.708884 | Stagnic Luvisol | 0 - 30 | 3 | 18 | 80 | 1.04 | 1.33 | 41.61 | 2009 | 2009 - 16 | SOC, DRIFTS, SMB-C |
| Kraichgau 3 | 48.927197 | 8.715891 | Stagnic Luvisol | 0 - 30 | 3 | 17 | 81 | 0.89 | 1.44 | 38.50 | 2009 | 2009 - 16 | SOC, DRIFTS, SMB-C |
| Swabian Jura 1 | 48.527510 | 9.769429 | Calcic Luvisol | 0 - 30 | 3 | 38 | 56 | 1.78 | 1.32 | 70.33 | 2009 | 2009 - 16 | SOC, DRIFTS, SMB-C |
| Swabian Jura 2 | 48.529857 | 9.773253 | Anthrosol | 0 - 30 | 3 | 29 | 68 | 1.95 | 1.38 | 80.85 | 2009 | 2009 - 13 | SOC, DRIFTS, SMB-C |
| Swabian Jura 3 | 48.547035 | 9.773176 | Rendzic Leptosol | 0 - 30 | 3 | 45 | 51 | 1.91 | 1.07 | 61.27 | 2009 | 2009 - 13 | SOC, DRIFTS, SMB-C |

UTM = Universal Transverse Mercator reference system, SOC= soil organic carbon, Rep.= replicates, SOC = soil organic carbon, DRIFTS = Diffuse reflectance mid infrared Fourier transform spectroscopy, SMB-C = soil microbial biomass carbon, [1] = Ultuna continuous soil organic matter field experiment (Kätterer et al., 2011), [2] = Bad Lauchstädt extreme farmyard manure experiment (Blair et al., 2006)

**Table 2 Values of the two DAISY parameter sets used in this study. The parameters consist of turnover rates (k), maintenance respiration (only for SMB, added to the turnover rate), carbon use efficiency (CUE -which divides between carbon assimilated by SMB and lost as CO₂), the humification efficiency (fSOM_slow) and microbial recycling (part of SMB going directly back to SMB fast at turnover of either SMB pool). A graphical display of the model structure and pools considered within this study is found in Figure 1.**

| Parameter | Default DAISY | Bruun (2003) | Unit |
|---|---|---|---|
| kSOM_slow | $2.70 * 10^{-6}$ [#] | $4.30 * 10^{-5}$ [x] | $d^{-1}$ |
| kSOM_fast | $1.40 * 10^{-4}$ [#] | $1.40 * 10^{-4}$ [#] | $d^{-1}$ |
| kSMB_slow | $1.85 * 10^{-4}$ [*] | $1.85 * 10^{-4}$ [*] | $d^{-1}$ |
| kSMB_fast | $1.00 * 10^{-2}$ [*] | $1.00 * 10^{-2}$ [*] | $d^{-1}$ |
| kAOM_slow | 0.012 [*] | 0.012 [*] | $d^{-1}$ |
| kAOM_fast | 0.05 [*] | 0.05 [*] | $d^{-1}$ |
| maint_SMB_slow | $1.80 * 10^{-3}$ [*] | $1.80 * 10^{-3}$ [*] | $d^{-1}$ |
| maint_SMB_fast | $1.00 * 10^{-2}$ [*] | $1.00 * 10^{-2}$ [*] | $d^{-1}$ |
| CUE_SMB | 0.60 [#] | 0.60 [#] | kg kg$^{-1}$ |
| CUE_SOM_slow | 0.40 [*] | 0.40 [*] | kg kg$^{-1}$ |
| CUE_SOM_fast | 0.50 [*] | 0.50 [*] | kg kg$^{-1}$ |
| CUE_AOM_slow | 0.13 [*] | 0.13 [*] | kg kg$^{-1}$ |
| CUE_AOM_fast | 0.69 [*] | 0.69 [*] | kg kg$^{-1}$ |
| fSOM_slow (humification efficiency) | 0.10 [#] | 0.30 [x] | kg kg$^{-1}$ |
| part. SMB > SOM_fast (microbial recycling) | 0.40 [#] | 0.40 [#] | kg kg$^{-1}$ |
| fraction of SOM_slow at steady state Bruun (2002) equation | 0.83 | 0.49 | kg kg$^{-1}$ |

k = turnover rate (=death rate for SMB), maint = maintenance respiration (SMB only), CUE = carbon use efficiency, SOM = soil organic matter pools, SMB = soil microbial biomass pools, AOM = added organic matter pools (not considered in this study), part. = partitioning; Source: [#] original Jensen (1997), [*] modified by Müller (1997), [x] modified by Bruun (2003)

**Table 3 Measured soil properties of the bare fallow experiments at each site corresponding to the start of the bare fallow experiment and the end of the simulated period. Measurements include SOC and SMB-C stocks in the modeled layer, and the percentage of SOC that would be assigned to the slow pool according to DRIFTS stability index (DSI) measured at 105°C.**

| Site | Start year of experiment | End year of simulation | Depth of modeled layer (cm) | Bulk density of modeled layer (Mg/m³) | SOC at start Mg/ha* | SOC at end Mg/ha* | SMB-C at start Mg/ha* | SMB-C at end Mg/ha* | % SOC in slow pool at start (DSI 105°C) | % SOC in slow pool at end (DSI 105°C) | % of initial SOC lost | Number of years | % of initial SOC lost per year |
|---|---|---|---|---|---|---|---|---|---|---|---|---|---|
| Ultuna | 1956 | 2005 | 0 - 20 | 1.44 | 43.22 | 26.51 | NA | NA | 54 | 91 | 39% | 50 | 0.8% |
| Bad Lauchstädt | 1983 | 2008 | 0 - 20 | 1.24 | 45.08 | 41.91 | NA | NA | 70 | 80 | 7% | 26 | 0.3% |
| Kraichgau 1 | 2009 | 2015 | 0 - 30 | 1.37 | 37.10 | 32.59 | 0.847 | 0.408 | 80 | 98 | 12% | 7 | 1.7% |
| Kraichgau 2 | 2009 | 2015 | 0 - 30 | 1.33 | 41.61 | 38.66 | 0.853 | 0.314 | 73 | 93 | 7% | 7 | 1.0% |
| Kraichgau 3 | 2009 | 2015 | 0 - 30 | 1.44 | 38.50 | 35.06 | 0.672 | 0.261 | 76 | 99 | 9% | 7 | 1.3% |
| Swabian Jura 1 | 2009 | 2015 | 0 - 30 | 1.32 | 70.33 | 63.29 | 1.566 | 0.654 | 64 | 83 | 10% | 7 | 1.4% |
| Swabian Jura 2 | 2009 | 2013 | 0 - 30 | 1.38 | 80.85 | 79.61 | 1.805 | 0.970 | 66 | 83 | 2% | 5 | 0.3% |
| Swabian Jura 3 | 2009 | 2013 | 0 - 30 | 1.07 | 61.27 | 70.29 | 1.350 | 0.990 | 61 | 76 | -15% | 5 | -2.9% |

SOC = soil organic carbon, SMB-C = soil microbial biomass carbon, DSI = Diffuse reflectance mid infrared Fourier transform spectroscopy stability index, NA = no data available for this site, *stocks in Mg/ha refer to stocks within the depth of the modelled layer

**Table 4. Effect of the initialization method on simulation errors. Displayed are estimated least square means of the absolute error of DAISY bare-fallow simulations for SOC and SMB-C for the sites of Ultuna, Bad Lauchstädt and Kraichgau + Swabian Jura combined. Means are the estimate for the end of the simulation period (number of years in brackets). Different capital letters indicate significant differences (p < 0.05) within columns (not tested between sites). For Bad Lauchstädt, the initialization effect was non-significant, so only the least square means for the effect of the parameter set is displayed.**

| Parameter set | Initialization method of SOM pools | Ultuna (50yr) Least square means of errors (SOC Mg/ha) | Bad Lauchstädt (23yr) Back transformed least square means of errors (SOC Mg/ha) | Kraichgau + Swabian Jura (7 yr) Back transformed least square means of errors (SOC Mg/ha) | Kraichgau + Swabian Jura (7 yr) Least square means of errors (SMB-C Mg/ha) |
|---|---|---|---|---|---|
| Mueller (1997) | ratio of steady state assumption | 13.91 [A] | | 4.50 [A] | 0.354 [A] |
| | band area ratio of DRIFTS at 32°C | 10.86 [B] | 2.22 [A] | 4.50 [A] | 0.317 [AB] |
| | band area ratio of DRIFTS at 65°C | 10.06 [C] | | 4.42 [A] | 0.274 [ABC] |
| | band area ratio of DRIFTS at 105°C | 8.52 [D] | | 4.28 [A] | 0.205 [CD] |
| Bruun (2003) | ratio of steady state assumption | 5.84 [H] | | 3.12 [B] | 0.231 [BCD] |
| | band area ratio of DRIFTS at 32°C | 7.06 [E] | 6.01 [B] | 3.31 [B] | 0.179 [CDE] |
| | band area ratio of DRIFTS at 65°C | 6.75 [F] | | 3.30 [B] | 0.160 [DE] |
| | band area ratio of DRIFTS at 105°C | 6.15 [G] | | 3.25 [B] | 0.131 [E] |

SOM = soil organic matter pools, SOC = soil organic carbon, SMB-C = soil microbial biomass carbon, DRIFTS = Diffuse reflectance mid infrared Fourier transform spectroscopy

**Table 5 Optimized turnover rates and humification efficiency of this study (using the combined site analysis with original weighting and DSI) compared to other Bayesian calibrations and standard values of commonly used models. Turnover rates of other models were normalized to the DAISY standard of 10°C using an exponential equation (exception Clifford et al. (2014), where no temperature was given).**

| Model | DAISY | ICBM | CBM-CFS3 | APSIM | own creation* | CENTURY | DAISY | DAISY |
|---|---|---|---|---|---|---|---|---|
| Reference | This study | Ahrens | Hararuk | Luo | Clifford | Parton | Mueller | Bruun |
| Year | 2019 | 2014 | 2017 | 2016 | 2014 | 1993 | 1997 | 2003 |
| Turnover rates of the fast pool at 10°C (d$^{-1}$) | | | | | | | | |
| minimum | $1.07 * 10^{-4}$ | $4.57 * 10^{-4}$ | $6.30 * 10^{-4}$ | | | | | |
| optimum | $2.29 * 10^{-4}$ | $4.57 * 10^{-3}$ | $1.97 * 10^{-4}$ | | | $9.32 * 10^{-5}$ | $1.40 * 10^{-4}$ | $1.40 * 10^{-4}$ |
| maximum | $3.27 * 10^{-4}$ | $2.28 * 10^{-2}$ | $1.05 * 10^{-3}$ | | | | | |
| Turnover rates of the slow pool at 10°C (d$^{-1}$) | | | | | | | | |
| minimum | $2.99 * 10^{-6}$ | $4.57 * 10^{-7}$ | $9.86 * 10^{-6}$ | $1.00 * 10^{-4}$ | $1.10 * 10^{-4}$ | | | |
| optimum | $3.25 * 10^{-5}$ | $2.28 * 10^{-5}$ | $1.10 * 10^{-5}$ | $3.00 * 10^{-4}$ | $1.67 * 10^{-4}$ | $2.10 * 10^{-6}$ | $2.70 * 10^{-6}$ | $4.30 * 10^{-5}$ |
| maximum | $6.14 * 10^{-5}$ | $4.57 * 10^{-5}$ | $1.32 * 10^{-5}$ | $6.00 * 10^{-4}$ | $2.19 * 10^{-4}$ | | | |
| Portion of fast to slow pool = humification efficiency (dimensionless) | | | | | | | | |
| minimum | 0.05 | 0.05 | | | | | | |
| optimum | 0.34 | 0.2 | | | | 0.3 | 0.1 | 0.3 |
| maximum | 0.35 | 0.35 | | | | | | |

References: (Ahrens et al., 2014; Bruun et al., 2003; Clifford et al., 2014; Hararuk et al., 2017; Luo et al., 2016; Mueller et al., 1997; Parton et al., 1993), * remarks – Clifford et al. (2014) did not specify a base temperature for their model

## 10    Figures

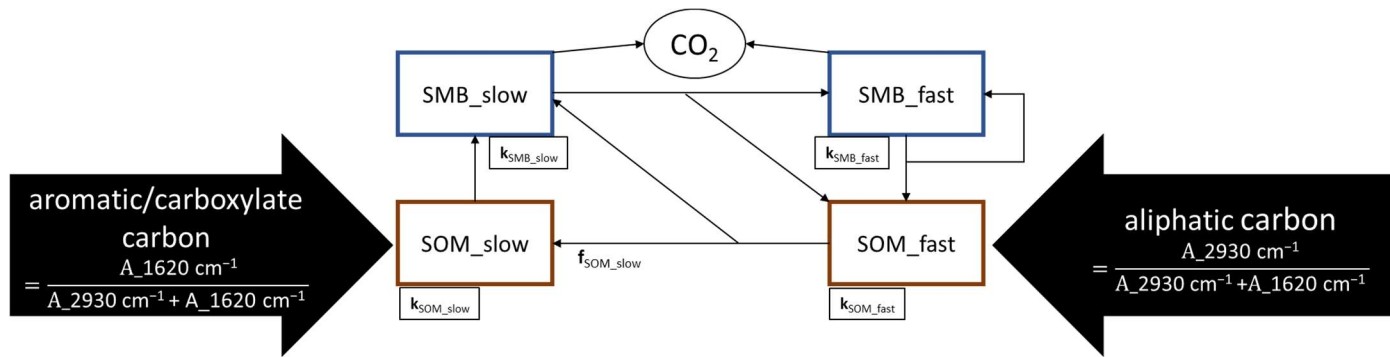

**Figure 1 Original structure of the internal cycling of SOM in the DAISY model, as it was used in this study. A_2930 cm$^{-1}$ and A_1620 cm$^{-1}$ refer to the areas below the DRIFTS absorption bands at 2930 cm$^{-1}$ and 1620 cm$^{-1}$ (Equation 3),** *kSOM* **and k***SMB (fast/slow)* **are turnover rates of the fast and slow SOM and SMB pools, respectively and** *fSOM_slow* **is the humification efficiency. All model parameters can be found in Table 2.**

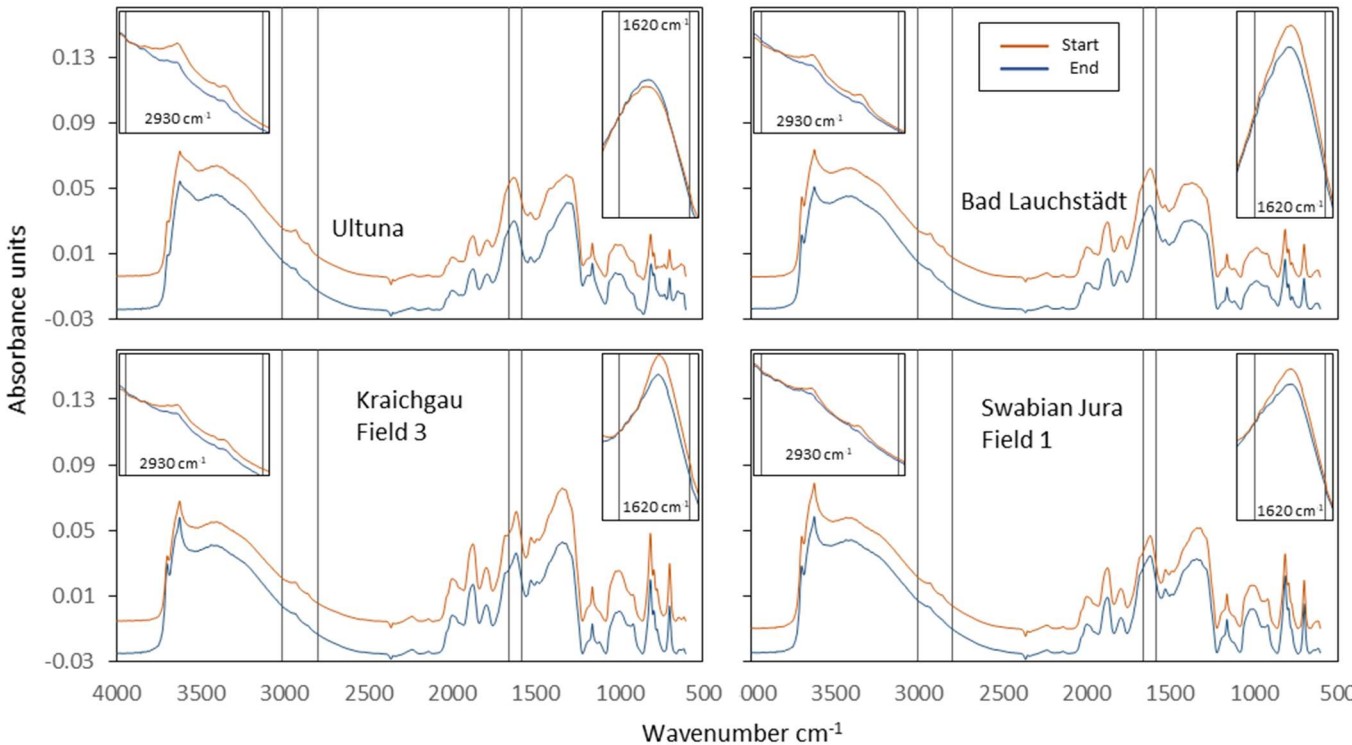

**Figure 2  Examples of baseline corrected and vector normalized DRIFTS spectra of bulk soil samples (dried at 105°C) of the first and last year of the bare fallow plots at four sites. Fallow periods were 50 years (Ultuna), 24 years (Bad Lauchstädt) and 7 years (Kraichgau and Swabian Jura). Small pictures on the top left and right, are zoomed in versions of the 2930cm$^{-1}$ band and the 1620cm$^{-1}$ band, respectively For better visibility, the full spectra pictures have a y-axis offset (+0.02 for samples from the start), while zoomed in versions share a common baseline. More details on the sites in Table 3.**

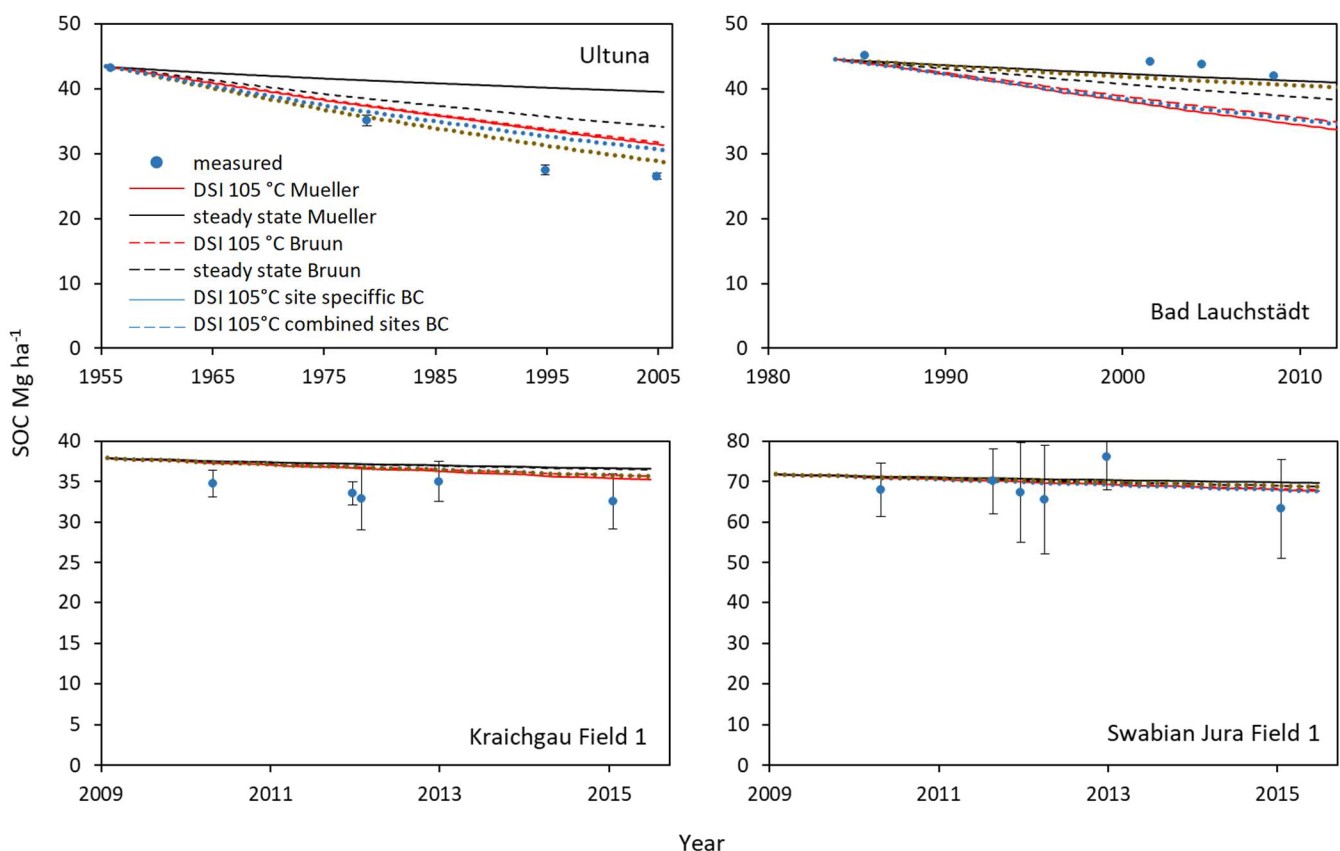

**Figure 3 Example of SOC simulations from Ultuna (top left), Bad Lauchstädt (top right), Kraichgau field 1 (bottom left) and Swabian Jura Field 1 (bottom right). Initializations were done (i) assuming steady state using the formula of Bruun and Jensen (2002) (equation 1) with both turnover rates of Mueller et al. (1997) and Bruun et al. (2003) and (ii) by the DRIFTS stability index (DSI) at 105°C drying temperature using both turnover rates for simulations (simulations using the other drying temperatures for DSI in the supplementary). The site specific and the combination of all sites Bayesian calibrations (BC) are also displayed. Bars indicate the standard deviation of measured values of all plots (n = 3) per field.**

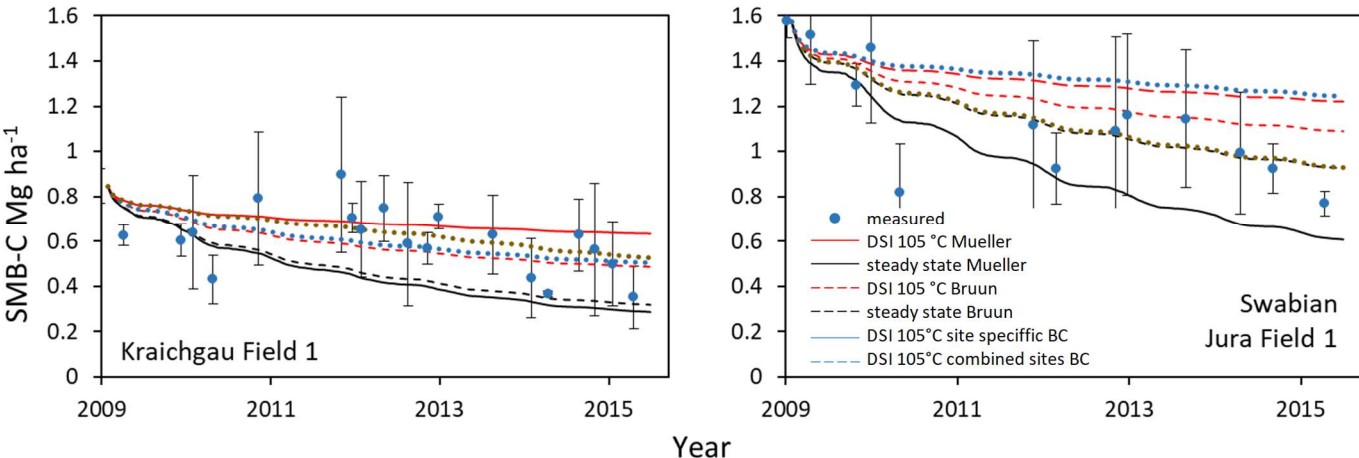

**Figure 4 Example SMB-C simulations for Kraichgau field 1 (left) and Swabian Jura Field 1 (right). Initializations were done (i) assuming steady state using the formula of Bruun and Jensen (2002) with turnover rates of Mueller et al. (1997) and Bruun et al. (2003) and (ii) by the DRIFTS stability index (DSI) at 105°C drying temperature using both turnover rates for simulations (simulations using the other drying temperatures for DRIFTS in the supplementary). The site specific and the combination of all sites Bayesian calibrations (BC) are also displayed. Bars indicate the standard deviation of measured values of all plots (n =3) per field.**

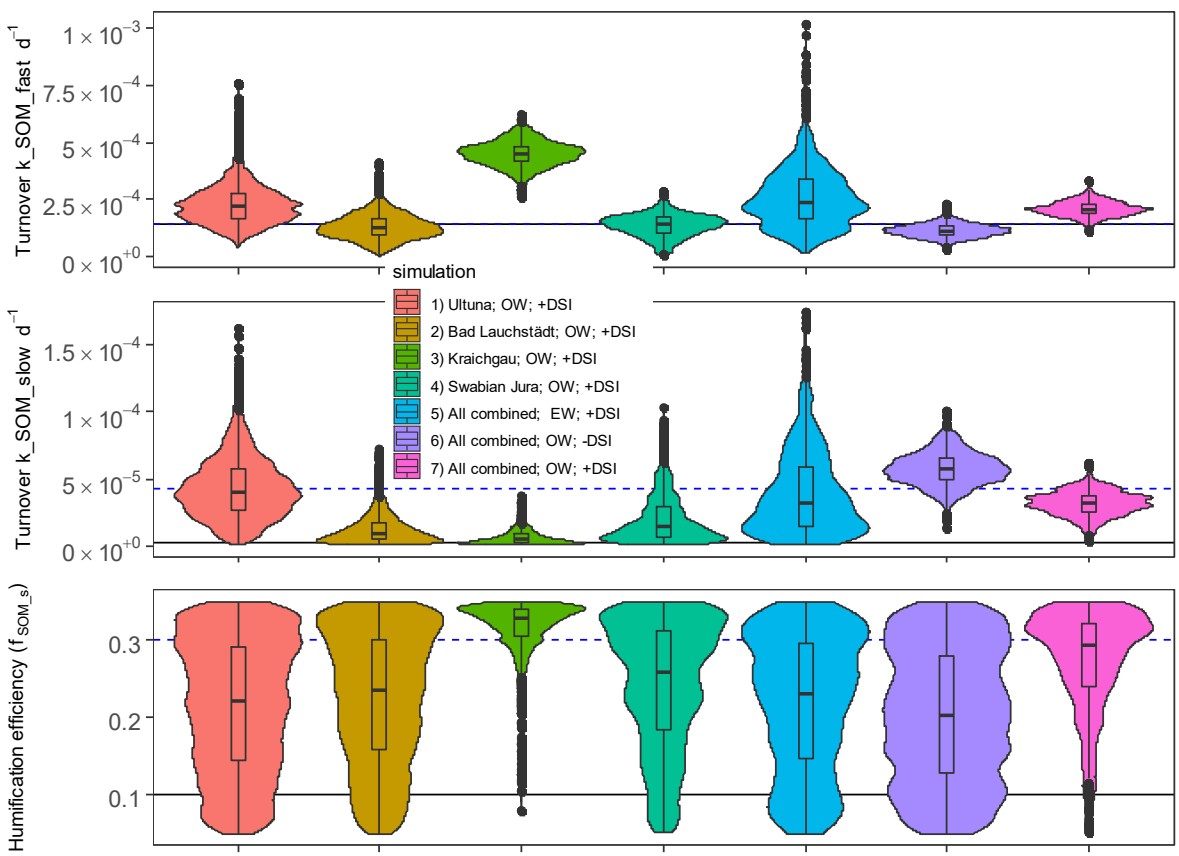

**Figure 5 Violin plots of the parameter distributions, obtained by the Bayesian calibration using only the individual sites (1-4) and all sites combined (5-7) with different weighing schemes (OW = original weight, EW = equal weight calibration; +/- DSI indicates, whether the DSI data was used for calibration). The black line corresponds to the parameters of Mueller (1997), the blue dashed line to the parameters of Bruun (2003). Note: The turnover *k_SOM_fast* parameter (top figure) is the same in both Mueller (1997) and Bruun (2003)**

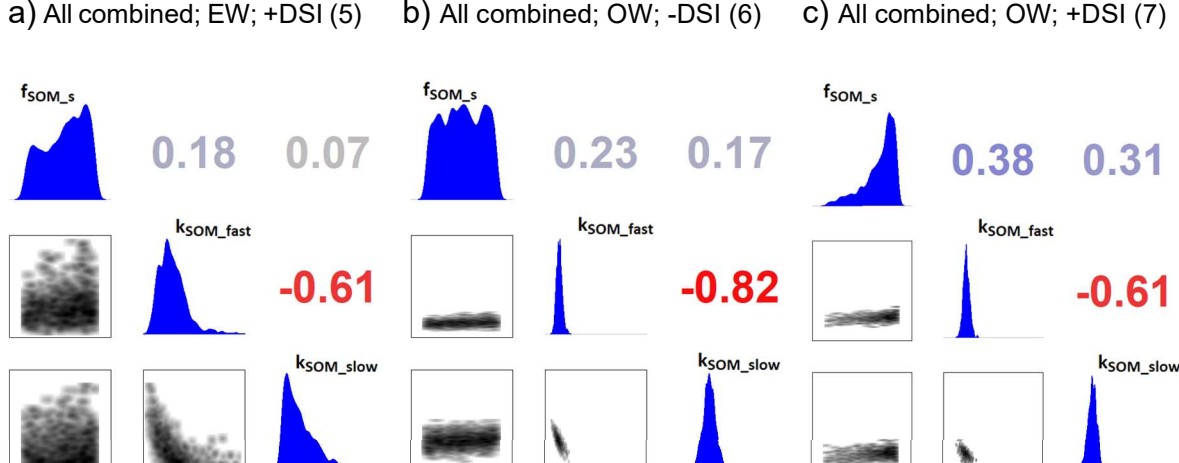

**Figure 6 Correlation matrices of posterior distributions from the Bayesian calibrations of a) equal weight calibration for all sites combined using DSI (5), b) original weight calibration for all sites combined without using DSI (6), and c) original weight calibration for all sites combined using the DSI (7). The plots of individual site simulations (1-4) can be found in the supplemental material.**

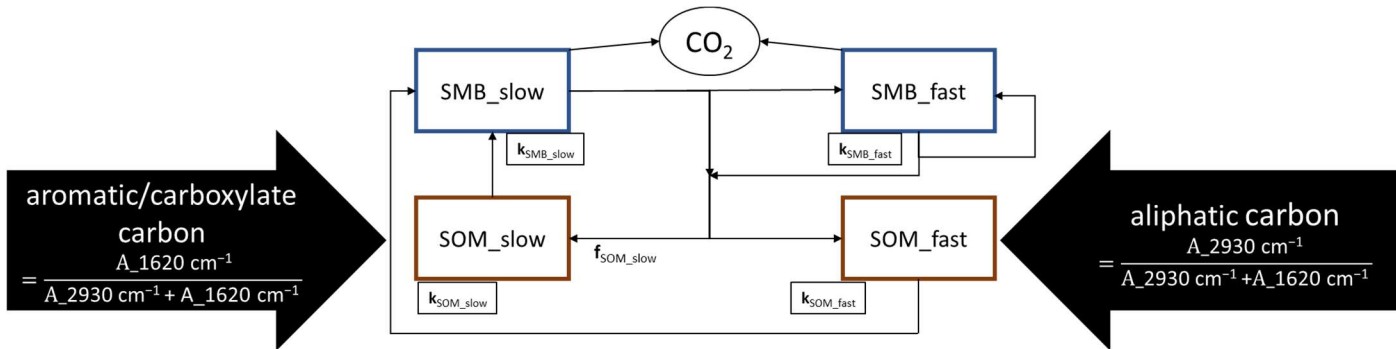

**Figure 7 Suggested improvements to the internal cycling structure of SOM in the DAISY model. The division into fast and slow cycling SOM, corresponding to aliphatic and aromatic/carboxylate carbon follows the turnover/death of either SMB pool. Aliphatic carbon no longer becomes aromatic/carboxylate carbon without the involvement of microbes.**