# Peer review of "DRIFTS band areas as measured pool size proxy to reduce parameter uncertainty of soil organic matter models"

_Biogeosciences, 2019_

## Referee Comment (RC1) · Sander Bruun (Referee) · 29 Aug 2019

General comments

The papers deals with initialization of pools in the soil organic matter model of Daisy. The paper is using unique datasets for long-term fallow treatments to test a new way of initializing the soil organic matter pools based on specific peaks in the DRIFTS specta of the soils. Pool initialization of SOM model is an important issue that is still causing some difficulties with the currently used approaches. The paper is therefore very timely and present an interesting approach that could be useful in many situations. The work is of a high quality and based on high quality data and the manuscripts is well written.

[Figure]

Specific comments

Line 78. I agree that the DSI can be better than the steady-state assumption, but perhaps it is worth discussing this in a little more detail. If information about the history of the site is available then that method should work. This require that the history is known for millennia, and that is rarely the case.

Line 118: Was soil samples from through out the experimental period analyzed? Please specify.

Line 129: The spectra were not recorded in absorbance, but subsequently converted to absorbance units, right?

Line 130: I wonder how much this way of determining the DSI is affected by the instrument i.e. if somebody took the same soils and did the measurement on another instrument would the get the same DSI and pool sizes. I am afraid that it would be quite much affected by that especially if you use other IR detection techniques. Maybe it would be worth addressing this in the discussion.

Line 181: It says 84% and not 83% in Table 2. Please correct where appropriate.

Line 196 to 209: I am not entirely sure I understand what the point of analyzing the SMEx with a statistical model is. I think you should consider whether it add enough understanding to warrant inclusion. Alternatively explain the point a little better.

Line 236-237. The necessity of constraints on the fSOM-Slow parameter is a little problematic. I cannot help thinking that it means that the data, which is used for calibration, is insufficient. With these restraints, I guess you are likely to end up with a value of 0.35 which is rather arbitrarily chosen by you.

Line 364-365. I agree that even though we have had the same management for a long time the steady-state assumption is not valid. However, I believe that the reason for this has to do with longer-term effects rather than the smaller effects that you mention i.e. variation in climate agricultural management. If you look at a longer terms, most

sites would probably have been deforested within the last 2000 years. Because of the high inputs from the forest, this could have resulted in an unusually large fraction of resistant organic matter that has not been degraded from that period. Also it is very common with drained soils soil. This means that the soil at some time it its history has had a very high water table and perhaps even been inundated. We know that this can result in significant accumulation of organic matter. After the soil has been drained, this has led to a large residual of resistant C again. The same could happen if there has been a history of fires with inputs of charcoal. Perhaps this is worth discussing a bit more.

Line 373: I cannot help it thinking that it is somewhat of a coincidence that you get better model performance with the DSI as long as you have not recalibrated the model. Of course using more data as for example DSI to restrain the model should improve the model, but only after it has been recalibrated.

It is not entirely clear what data were used for the calibrations based on DSI. As far as I understand, you measured DSI of all the soil samples and that means that you can compare the simulated distribution between fast_SOM1 and slow_SOM with the one measured and calculated using formula (2) and a similar formula for fast_SOM. Is this the case? And if it is why have you not shown the "measured" value of fast and slow SOM and compared it with the modelled?

Is it worth publishing the optimal parameters selected by the Baysian calibration based on DSI?

---

## Referee Comment (RC2) · Anonymous Referee #2 · 30 Aug 2019

**General comments**

Laub and Colleagues present interesting ideas how DRIFTS spectra could be used to initialize and calibrate soil organic matter models. What warrants more discussion is that with their results we should put again more emphasis on the chemical recalcitrance hypothesis, i.e. that molecular properties determine the persistence of organic matter in soils. The literature seems to disagree (Schmidt et al., 2011). If we indeed assign the aromatic peak to slow cycling pools with a turnover time of 426 years and the aliphatic peak to a fast cycling pool with 47 to 90 years, the authors would contradict the synthesis of Schmidt et al. (2011) (their Figure 1, for example). In my opinion, it

would be interesting if the authors could at least discuss how their DRIFTS peaks could be useful for the new class of microbial-mineral models such as Tang and Riley (2015) or Sulman et al., (2014).

Specific comments

The authors state that "the DRIFTS initialization of SOM pools significantly reduced model errors of poor performing model runs assuming steady state, irrespective of the turnover rates used, but the faster turnover parameter set fit better to all sites except Bad Lauchstädt. This suggests that soils under long-term agricultural use were not necessarily at steady state." In my opinion this statement is not backed up by their results. The Bruun parameters with steady state assumption perform better at Ultuna and Kraichgau + Swabian Jura (Table 4) for SOC stocks.

The authors also state that "[. . .] two approaches [. . .] significantly reduced parameter uncertainty and equifinality". One of the approaches was the inclusion of DRIFTS. But looking at the violin plots in Figure 5, only the humification efficiency seems to be better constrained. I suggest modifying the statement towards this direction.

I agree with the other reviewer, Sander Bruun, that analyzing the squared model errors with a statistical model should at least be better explained.

The manuscript would benefit from a thorough spell and language check.

Schmidt MWI, Torn MS, Abiven S et al. (2011) Persistence of soil organic matter as an ecosystem property. Nature, 478, 49-56.

Sulman BN, Phillips RP, Oishi AC, Shevliakova E, Pacala SW (2014) Microbe-driven turnover offsets mineral-mediated storage of soil carbon under elevated CO2. Nature Clim. Change, 4, 1099-1102.

Tang J, Riley WJ (2015) Weaker soil carbon-climate feedbacks resulting from microbial and abiotic interactions. Nature Climate Change, 5, 56-60.

---

## Referee Comment (RC3) · Lauric Cécillon (Referee) · 31 Aug 2019

"Reservation on the rationale of the DRIFTS stability index of soil organic matter (SOM) in mineral soil, and its use for partitioning the C kinetic pools of SOM dynamics models"

This draft by Laub and colleagues describes a method to divide soil organic matter (SOM) into fast and slow cycling C pools in the soil organic module of the DAISY model. This method is based on the characterization of bulk mineral soil samples using mid-infrared diffuse reflectance spectroscopy (DRIFTS). DRIFTS spectra of bulk mineral soils are used to compute the "DRIFTS stability index" of SOM, defined as the ratio of aliphatic C-H (2930 cm-1) to aromatic C=C (1620 cm-1) stretching vibrations.

[Figure]

The DRIFTS stability index was previously published by Demyan et al. (2012) in the European Journal of Soil Science.

The development of routine and operational method to initialize the relative size of C kinetic pools from SOM dynamics models is a very important and timely topic. Indeed, the accuracy of the simulations of SOM evolution in mineral soils by current models is strongly questioned, notably because a poor initialization of the size of C kinetic pools.

The method proposed by Laub and colleagues, using the DRIFTS stability index to divide soil organic matter (SOM) into fast and slow cycling C pools in the soil organic module of the DAISY model is original and very interesting, and their draft is well structured and written.

However, I have a major concern regarding the rationale of the DRIFTS stability index of SOM in mineral soil, and its use for partitioning the C kinetic pools of SOM dynamics models. In this review, I will only discuss this concern, though this stimulating and timely work would deserve many other comments, as highlighted by the two other reviewers of this draft.

First, I would like to come back on the justification of the DRIFTS stability index by Demyan and colleagues in their 2012 paper. Demyan et al. (2012) searched for information related to SOM in DRIFTS spectra of bulk mineral soils, and its link to SOM stability as assessed by a SOM density fractionation scheme. In their search for SOM information in DRIFTS spectra of bulk mineral soils, they discarded "wavenumbers of functional groups associated with non-organic compounds such as silicates and alumino-iron oxides". For them, "these criteria removed the peaks <1000 cm−1 and the peaks at 1980, 1870, 1792 and 1390 cm−1", but not the 1620 cm−1 peak. For them, "the [DRIFTS] peak at 1620 cm−1 was assigned to predominately aromatic C = C stretching and/or asymmetric–COO− stretching but possibly also C = O vibrations".

Demyan et al. (2012) show that "a positive relationship was found between the ratio of the peaks at 1620 and 2930 cm−1 (1620:2930) and the ratio of stable C (sum of

C contained in clay and >1.8 g cm−3 fractions) to labile C (amount of C in the <1.8 g cm−3 fraction) ($R^2$ = 0.62, P = 0.012)." For the authors, this result justifies that the DRIFTS stability index can reliably be "taken as an indicator of SOM stability" (Demyan et al., 2012).

However, a short look at the literature on DRIFTS of soils show that the 1620 cm-1 peak in bulk mineral soils cannot be exclusively assigned to absorption from SOM functional groups (C = C or C = O) as claimed by Demyan et al. 2012. I will only cite two important papers: Nguyen et al. (1991) and Reeves (2012).

Nguyen and colleagues, based on DRIFTS spectra of pure mineral compounds and various soil samples demonstrated that "The DRIFT spectra of soils containing organic matter show considerable overlap of the silicate combination bands in the 2000-1600 cm-1 region". I provide here the Figure 1 modified from Nguyen et al. (1991) showing the DRIFTS spectra of quartz (pure or diluted in KBr), highlighting the strong absorption of quartz at 1620 cm-1 (for the DRIFT spectra of pure quartz). They suggested that "Spectral subtraction techniques or prior chemical treatment may thus be required to resolve these peaks." (Nguyen et al., 1991). Reeves (2012) based on works similar than Nguyen et al. (1991), concluded that "With the exception of the bands at 2930 and 2850 cm−1 due to aliphatic CH [when the soil does not contain carbonates, added by me] and the large OH band spanning most of the region between 2700 and 3500 cm−1, there is little that is obviously due to OM in the soil spectra". Regarding the 1620 cm-1 DRIFTS peak, he suggested, following Nguyen et al. (1991) that "the region between 1750–1600 cm−1 can be interpreted, despite the presence of strong silica bands, because silica can be ash subtracted quite well". But he also concluded his paper with this warning regarding spectral subtraction: "It will detect not only whether your sample is changed by 0.1% at some point in time, but will also seem to detect the phases of the moon and the mood you were in while you were measuring the data." (Hirschfeld, 1984; cited by Reeves, 2012).

I deduce from this short literature survey that in their 2012 paper, Demyan et al. incor-

rectly assigned to SOM compounds (C = C, C = O) exclusively the 1620 cm-1 DRIFTS peak of bulk mineral soils, as this peak is also due to mineral compounds such as quartz (but also to water in some phyllosilicates).

To further illustrate how the 1620 cm-1 DRIFTS peak of bulk mineral soil is poorly related to SOM compounds, I provide the Figure 2 based on published and unpublished data from the paper of Barré et al. (2016) in Biogeochemistry showing the non-parametric Spearman's Rho coefficient of DRIFTS spectra from soils coming from the Ultuna Fame trial, one site that was used in this reviewed work by Laub and colleagues, with SOC concentration. In Figure 2, we clearly see the strong and positive Rho coefficient of the 2900 cm-1 spectral region with SOC concentration while the 1620 cm-1 spectral region show a Rho coefficient with SOC concentration close to 0, suggesting (though not demonstrating) that other compounds that organic matter absorb energy in the 1620 cm-1 spectral region of DRIFTS spectra, when scanning bulk mineral soils.

From the above-mentioned information, I therefore question the rationale of the DRIFTS stability index of soil organic matter (SOM) in mineral soil samples.

My interpretation is that this index is dividing a quantity that is highly correlated to SOC concentration (the 2900 cm-1 spectral region), by a quantity that is weakly changing when SOC concentration is modified (the 1620 cm-1 spectral region, provided a similar mineral composition). The DRIFTS stability index may thus show an increased SOC lability when SOC concentration is increased. I thus hypothesize that the DRIFTS stability index, as proposed by Demyan et al. (2012) and Laub and colleagues in this reviewed draft, may provide some information that is basically the same (though with added noise) than a variable much simpler than their index: total SOC concentration. It is well documented that an increase in SOC concentration is associated with an increased in the labile/stable SOC ratio, and all proposed indicators of SOM stability should be compared to SOC concentration, the most simple and straightforward indicator of SOM stability (though not very accurate). What is the Spearman's Rho coefficient of the DRIFTS stability index with SOC concentration in the dataset of Laub

and colleagues?

I suggest that the authors (rather than using the spectral subtraction technique suggested by Nguyen et al., 1991 or Reeves, 2012), (i) test a soil dilution in KBr to reduce mineral artifacts in the 1620 cm-1 spectral region of neat DRIFTS, (ii) or test attenuated total reflectance mid-infrared spectroscopy (MIR-ATR) as an alternative technique. Inded, MIR-ATR is a technique where the 1620 cm-1 peak region seems to be much less affected by quartz and other minerals that neat DRIFT signal, as illustrated in Figure 3 (Cécillon, Unpublished data).

Finally, as Laub and colleagues benefit from soil samples from two long-term bare fallow sites in Europe, I suggest that they compute the Spearman's Rho coefficient of their DRIFTS stability index with the proportion of centennially persistent soil organic carbon (CPsoc), that may be derived from the SOC evolution in the bare fallow plots, as shown by Cécillon et al. (2018). A higher Spearman's rho coefficient of the DRIFTS stability index with CPsoc than the Spearman's rho coefficient of SOC concentration with CPsoc, would suggest an added value of the index compared to SOC concentration, in its current state.

References:

Barré, P., Plante, A. F., Cécillon, L., Lutfalla, S., Baudin, F., Bernard, S., Christensen, B. T., Eglin, T., Fernandez, J. M., Houot, S., Kätterer, T., Le Guillou, C., Macdonald, A., van Oort, F., and Chenu, C. (2016). The energetic and chemical signatures of persistent soil organic matter, Biogeochemistry, 130, 1-12.

Cécillon, L., Baudin, F., Chenu, C., Houot, S., Jolivet, R., Kätterer, T., Lutfalla, S., Macdonald, A., van Oort, F., Plante, A. F., Savignac, F., Soucémarianadin, L. N., and Barré, P. (2018). A model based on Rock-Eval thermal analysis to quantify the size of the centennially persistent organic carbon pool in temperate soils, Biogeosciences, 15, 2835-2849.

Demyan, M. S., Rasche, F., Schulz, E., Breulmann, M., Müller, T. and Cadisch, G. (2012). Use of specific peaks obtained by diffuse reflectance Fourier transform mid-infrared spectroscopy to study the composition of organic matter in a Haplic Chernozem, European Journal of Soil Science, 63, 189-199.

Nguyen, T.T., Janik, L.J. & Raupach, M. (1991). Diffuse reflectance infrared Fourier transform (DRIFT) spectroscopy in soil studies, Australian Journal of Soil Research, 29, 49–67.

Reeves JB (2012). Mid-infrared spectral interpretation of soils: is it practical or accurate?, Geoderma 189, 508-513.

[Figure]

**Infrared spectra of powdered quartz**
A: neat DRIFT; B: 3% in KBr DRIFT; C: 0.3% in KBr disc
(Figure modified from Nguyen et al., 1991)

**Fig. 1.**

[Figure]

**Spearman's Rho of DRIFTS spectra with SOC concentration at Ultuna Frame trial (n=34)**
Information on soil samples & DRIFTS data acquisition and treatment is described in Barré et al., 2016.

**Fig. 2.**

[Figure]

Fig. 3.

---

## Author Comment (AC1) · 9 Oct 2019

We would like to thank the editor for taking the time to handle our manuscript and for finding three very constructive reviewers. We also want to thank all reviewers for taking the time and reviewing our manuscript to help improve its quality. We are grateful for the honest and thorough feedback. The suggestions were highly useful and provided us with information, where misunderstandings could be possible and where we needed to make our message clearer and to discuss the limitations of the DSI in more detail. They helped to further improve the quality of this manuscript and we hope that we addressed concerns to a satisfying extent. Our comments to the reviewers in the following are in blue color. We made use of the constructive criticism and altered the text of the manuscript, where applicable. We added screenshots of alterations in the text related to the comments. These are displayed in green color.

**Comments of Reviewer 1: Sander Bruun**

**General comments**

The papers deals with initialization of pools in the soil organic matter model of Daisy. The paper is using unique datasets for long-term fallow treatments to test a new way of initializing the soil organic matter pools based on specific peaks in the DRIFTS specta of the soils. Pool initialization of SOM model is an important issue that is still causing some difficulties with the currently used approaches. The paper is therefore very timely and present an interesting approach that could be useful in many situations. The work is of a high quality and based on high quality data and the manuscripts is well written.

We are glad that reviewer found our work useful and interesting.

**Specific comments**

Line 78. I agree that the DSI can be better than the steady-state assumption, but perhaps it is worth discussing this in a little more detail. If information about the history of the site is available then that method should work. This require that the history is known for millennia, and that is rarely the case.

We agree and added a corresponding sentence as suggested, at line 56.

> lack of data of residue input and weather data for the required long-term timescales (from > 200 years to millennia). So, while the approach should work in theory, the history of a site is usually not known for the
> 60 timescales that SOM needs to equilibrize. Therefore, the simulation of past carbon inputs and the assumption of

Line 118: Was soil samples from throughout the experimental period analyzed? Please specify.

Yes, from throughout the period. We specified this now more clearly.

> Bulk soil samples from the start and throughout the simulation period of all experiments were analyzed for total carbon and DRIFTS spectra; samples from the Kraichgau and Swabian Jura sites were additionally analysed for soil microbial biomass carbon (SMB-C). After sampling, all bulk soil samples (except for SMB-C) were passed

Line 129: The spectra were not recorded in absorbance, but subsequently converted to absorbance units, right?

Yes – the wording was changed

combination of 16 co-added scans with a resolution 4 cm$^{-1}$. Spectra were recorded and then converted to absorbance units (AU); the acquisition mode "double-sided, forward-backward" and the apodization function

Line 130: I wonder how much this way of determining the DSI is affected by the instrument i.e. if somebody took the same soils and did the measurement on another instrument would the get the same DSI and pool sizes. I am afraid that it would be quite much affected by that especially if you use other IR detection techniques. Maybe it would be worth addressing this in the discussion.

Indeed, at least to our experience, there are some differences between the spectra of different spectrometers, especially between detectors. We added a sentence addressing this in chapter 4.1. As we already tested different temperatures for drying, which we found to be the most dominant factor affecting DSI, it was beyond the scope of this publication to test the effect of the spectrometer. We were first and foremost interested in, whether the DSI approach adds value in general to SOM initialization, which we think it does.

| 435 | There are some remaining questions that should be answered to standardize the application of the DSI for model initialization. Those are related to how the type of spectrometer influences the spectra, as well as how water and mineral interferences (Nguyen et al., 1991) in the spectra can be eliminated or at least be further reduced. We had the experience, that spectra and therefore peak areas vary to some degree between different spectrometers (mostly due to different detectors, types of detector cooling and resolution). Hence, it will be necessary to either |
| 440 | use the same spectrometer, or to develop techniques to standardize spectra across a large number of instruments. |

Line 181: It says 84% and not 83% in Table 2. Please correct where appropriate.

We have done so

Line 196 to 209: I am not entirely sure I understand what the point of analyzing the SMEx with a statistical model is. I think you should consider whether it add enough understanding to warrant inclusion. Alternatively explain the point a little better.

We wanted an analysis of the model error which could give us a better measure of model uncertainty, and since in some experiments (Swabian Jura and Kraichgau) we had several fields, make use of the statistical power provided by the experimental design. The second advantage of a statistical analysis of model error was, that we could analyze for a time trend (increase with time) of the model error.

| 200 | mixed model with $SME_x$ as response was then used to test for significant differences between initialization methods. This approach allowed us to make use of the statistical power of the three Kraichgau and Swabian Jura fields to analyze which initialization was most sound and for a trend of the model error with increasing simulation time. In some cases, $SME_x$ was transformed to ensure a normal distribution of residuals (square root |

Line 236-237. The necessity of constraints on the fSOM-Slow parameter is a little problematic. I cannot help thinking that it means that the data, which is used for calibration, is insufficient. With these restraints, I guess you are likely to end up with a value of 0.35 which is rather arbitrarily chosen by you.

From our perspective, rather than a data limitation, this is an indicator how model structure affects the results of Bayesian calibration. In the initial first Bayesian calibration without limits, fSOM_slow was well constrained by the calibration (Figure S5 in the manuscript), but to a value we consider

implausible (~ 95%). Therefore, we suggested a possible alternative formulation of DAISY (Figure 7 in original text). While recently testing the proposed revised model structure of DAISY, we found that with this new model formulation, fSOM_slow does not have a trend towards the upper constraints ( >= 80%) anymore (the high humification efficiency values here, are because little new SOM is coming in within the bare plots) , even without artificial constraints. See as an example the results of the new structure with (2) and without (3) the fSOM_slow constraints compared to (1) the initial BC of this study:

[Figure]

**Figure S 1 Violin plots of the parameters, obtained by the Bayesian calibration using the new suggested model structure (Old constraints are 0.05 and 0.35, no constraints means 0.01 and 0.99.). The black line corresponds to the parameters of Mueller (1997), the blue dashed line to the parameters of Bruun (2003).**

We also added two more sentences to discuss this points.

> 500    SOM aligned with the earlier published rates. If a parameter is problematic, such as *fSOM_slow* it could mean that there is a lack of data, especially if it is not identifiable by the Bayesian calibration. However, if parameters are clearly constrained by the Bayesian calibration, but those constraints are implausible, it usually means that the model structure is suboptimal (Poeter et al., 2005) and should be altered.

Line 364-365. I agree that even though we have had the same management for a longtime the steady-state assumption is not valid. However, I believe that the reason for this has to do with longer-term effects rather than the smaller effects that you mention i.e. variation in climate agricultural management. If you look at a longer terms, most sites would probably have been deforested within the last 2000 years. Because of the high inputs from the forest, this could have resulted in an unusually large fraction of resistant organic matter that has not been degraded from that period. Also it is very common with drained soils soil. This means that the soil at some time it its history has had a very high water table and perhaps even been inundated. We know that this can result in significant accumulation of organic matter. After the soil has been drained, this has led to a

large residual of resistant C again. The same could happen if there has been a history of fires with inputs of charcoal. Perhaps this is worth discussing a bit more.

We agree and added these possibilities to the main text.

> particular field. This is particularly relevant, given that the changes in genotypes of crops, agricultural
> management, crop rotations and the rise of average temperatures in recent decades as well as stronger land use
> 370  changes such as organic soils draining or deforestation in recent centuries probably have affected the past quality
> and quantity of carbon inputs to soil. Consequently, the steady state assumption for model initialization is not

Line 373: I cannot help it thinking that it is somewhat of a coincidence that you get better model performance with the DSI as long as you have not recalibrated the model. Of course using more data as for example DSI to restrain the model should improve the model, but only after it has been recalibrated.

We interpreted this from the fact that SMB-C simulations were best when using the DSI as indicator, even if the turnover rates are unclear. As SMB-C is a much faster reacting pool than TOC, which did not change that much in our trials in Kraichgau and Swabian Jura. The DSI at 105°C was consistently lower in model error for simulated SMB-C than the steady state initialization, which should indicate that it is a useful proxy regardless of turnover rate, as long as there is a clear distinction between fast and slow pools.

It is not entirely clear what data were used for the calibrations based on DSI. As far as I understand, you measured DSI of all the soil samples and that means that you can compare the simulated distribution between fast_SOM1 and slow_SOM with the one measured and calculated using formula (2) and a similar formula for fast_SOM. Is this the case? And if it is why have you not shown the "measured" value of fast and slow SOM and compared it with the modelled?

You are correct, we used the measured DSI throughout the simulation period for the Bayesian calibration. We are happy to provide the modelled vs measured DSI throughout the simulation period – we also added it to the manuscript:

> 300  The resulting amount of SOC in the slow pool according to the computed DSI changed from the initial range of
> 54 to 80 % to the range of 76 to 99% at the end of the observational period (Figure S 7). The SMB-C reacted

[Figure]

**Figure S 2 Development of simulated vs observed SOM in the slow pool, according to DSI division throughout the simulation period (for brevity only for 105 °C). Bars indicate standard deviation of all plots per field.**

Is it worth publishing the optimal parameters selected by the Baysian calibration based on DSI?

While we think that the ideal way to use our results is using the posterior probability distributions of our parameters, we have mentioned the parameter set of the maximum likelihood from our

Bayesian calibration in chapter 3.3 (0.34, $2.29 * 10^{-4}$, $3.25 * 10^{-5}$ for the original weight calibration and 0.06, $9.58 * 10^{-5}$ and $5.54 * 10^{-5}$ for the calibration using original weights and no DSI) and in Table 5.

---

## Author Comment (AC2) · 9 Oct 2019

We would like to thank the editor for taking the time to handle our manuscript and for finding three very constructive reviewers. We also want to thank all reviewers for taking the time and reviewing our manuscript to help improve its quality. We are grateful for the honest and thorough feedback. The suggestions were highly useful and provided us with information, where misunderstandings could be possible and where we needed to make our message clearer and to discuss the limitations of the DSI in more detail. They helped to further improve the quality of this manuscript and we hope that we addressed concerns to a satisfying extent. Our comments to the reviewers in the following are in blue color. We made use of the constructive criticism and altered the text of the manuscript, where applicable. We added screenshots of alterations in the text related to the comments. These are displayed in green color.

**Comments of Reviewer 2:**

**General comments**

Laub and Colleagues present interesting ideas how DRIFTS spectra could be used to initialize and calibrate soil organic matter models. What warrants more discussion is that with their results we should put again more emphasis on the chemical recalcitrance hypothesis, i.e. that molecular properties determine the persistence of organic matter in soils. The literature seems to disagree (Schmidt et al., 2011). If we indeed assign the aromatic peak to slow cycling pools with a turnover time of 426 years and the aliphatic peak to a fast cycling pool with 47 to 90 years, the authors would contradict the synthesis of Schmidt et al. (2011) (their Figure 1, for example).

We do not think, that our results contradict Schmidt et al. (2011). Rather, the DSI seems to point towards the same direction as other measures of SOM quality, such as the amount of SOM in different aggregate sizes and density fractions. This was actually shown in our previous works (Demyan et al., 2012). We have to keep in mind that the DSI is still only a proxy and dividing the whole continuum of SOM quality into two discrete "qualities" is a strong simplification of the real world. However, we think it seems to be a valid one, especially when two pool SOM models are to be used, which anyway divide SOM into two pools. Additionally, a physical protection of SOM is implicitly included in DAISY, in the form of a clay function reducing SOC turnover.

> 65°C > 32°C drying temperature (differences being sometimes but not always significant). It has to be noted,
> 420   that the DSI is not purely related to chemical recalcitrance of SOM, as it also correlates with the level of SOC
> protected by aggregation (Demyan et al., 2012). Hence, it is likely that aggregation and chemical recalcitrance
> are related.

In my opinion, it would be interesting if the authors could at least discuss how their DRIFTS peaks could be useful for the new class of microbial-mineral models such as Tang and Riley (2015) or Sulman et al., (2014)

We think that DRIFTS could also be useful for those models, because of a good correlation of the DSI to size density fractions (Demyan et al., 2012), which is thought more representative of structural protection mechanisms. We added one sentence about this in the discussion.

> physically meaningful and to reduce model uncertainty and equifinality. As the DSI also had a good correlation
> 565   with structurally protected SOM (Demyan et al., 2012) it could also fit very well to models that directly simulate
> the protection of SOM as a function of microbial activity (Sulman et al., 2014). A better understanding and the

**Specific comments**

The authors state that "the DRIFTS initialization of SOM pools significantly reduced model errors of poor performing model runs assuming steady state, irrespective of the turnover rates used, but the faster turnover parameter set fit better to all sites except Bad Lauchstädt. This suggests that soils under long-term agricultural use were not necessarily at steady state." In my opinion this statement is not backed up by their results. The Bruun parameters with steady state assumption perform better at Ultuna and Kraichgau + Swabian Jura (Table 4) for SOC stocks.

We agree that our original text could be misinterpreted, so we altered the wording. For this statement we placed more weight on the Kraichgau + Swabian Jura sites (because those consisted of six fields) and assumed that Ultuna with Bruun turnover rates were not performing poorly. We observed there a significant improvement of the estimation of the sensitive SMB-C pool, for both turnover rates, while for SOC stocks there were only significant differences in model errors between turnover rates but not between initialization methods.

efficiency, respectively). The DRIFTS initialization of SOM pools could significantly reduced model errors of poor performing model runs assuming steady state if they were performing poorly, irrespective of the turnover rates used, but the faster turnover parameter set fit better to all sites except Bad Lauchstädt. This suggests that

The authors also state that "[...] two approaches [...] significantly reduced parameter uncertainty and equifinality". One of the approaches was the inclusion of DRIFTS. But looking at the violin plots in Figure 5, only the humification efficiency seems to be better constrained. I suggest modifying the statement towards this direction.

It is true, that humification efficiency was the parameter most seriously constrained by the DSI, also the turnover of the slow carbon pool was more strongly constrained (standard deviation of $9.3 * 10^{-6}$ with DSI vs $12.3 * 10^{-6}$ without DSI). We altered the wording in the sentence, to be more accurate.

individually and for a combination of all sites. The two approaches which significantly reduced parameter uncertainty and equifinality were: 1) the addition of the physico-chemically based DRIFTS stability index (strongest for humification, but also for slow SOM), and 2) combining several sites into one Bayesian calibration, as derived turnover rates can be strongly site specific. The combination of all four sites showed that

I agree with the other reviewer, Sander Bruun, that analyzing the squared model errors with a statistical model should at least be better explained.

This was done, see the comment to Sander Bruun.

200    mixed model with $SME_x$ as response was then used to test for significant differences between initialization methods. This approach allowed us to make use of the statistical power of the three Kraichgau and Swabian Jura fields to analyze which initialization was most sound and for a trend of the model error with increasing simulation time. In some cases, $SME_x$ was transformed to ensure a normal distribution of residuals (square root

The manuscript would benefit from a thorough spell and language check.

This will be done on the final reviewed version manuscript.

**References:**

Demyan, M. S., Rasche, F., Schulz, E., Breulmann, M., Müller, T. and Cadisch, G.: Use of specific peaks obtained by diffuse reflectance Fourier transform mid-infrared spectroscopy to study the composition of organic matter in a Haplic Chernozem, Eur. J. Soil Sci., 63(2), 189–199, doi:10.1111/j.1365-2389.2011.01420.x, 2012.

---

## Author Comment (AC3) · 9 Oct 2019

We would like to thank the editor for taking the time to handle our manuscript and for finding three very constructive reviewers. We also want to thank all reviewers for taking the time and reviewing our manuscript to help improve its quality. We are grateful for the honest and thorough feedback. The suggestions were highly useful and provided us with information, where misunderstandings could be possible and where we needed to make our message clearer and to discuss the limitations of the DSI in more detail. They helped to further improve the quality of this manuscript and we hope that we addressed concerns to a satisfying extent. Our comments to the reviewers in the following are in blue color. We made use of the constructive criticism and altered the text of the manuscript, where applicable. We added screenshots of alterations in the text related to the comments. These are displayed in green color.

**Comments of Reviewer 3: Lauric Cécillon**

"Reservation on the rationale of the DRIFTS stability index of soil organic matter (SOM)in mineral soil, and its use for partitioning the C kinetic pools of SOM dynamics models" This draft by Laub and colleagues describes a method to divide soil organic matter(SOM) into fast and slow cycling C pools in the soil organic module of the DAISY model. This method is based on the characterization of bulk mineral soil samples using mid-infrared diffuse reflectance spectroscopy (DRIFTS). DRIFTS spectra of bulk mineral soils are used to compute the "DRIFTS stability index" of SOM, defined as the ratio of aliphatic C-H (2930 cm-1) to aromatic C=C (1620 cm-1) stretching vibrations.

The DRIFTS stability index was previously published by Demyan et al. (2012) in the European Journal of Soil Science.

The development of routine and operational method to initialize the relative size of C kinetic pools from SOM dynamics models is a very important and timely topic. Indeed, the accuracy of the simulations of SOM evolution in mineral soils by current models is strongly questioned, notably because a poor initialization of the size of C kinetic pools. The method proposed by Laub and colleagues, using the DRIFTS stability index to divide soil organic matter (SOM) into fast and slow cycling C pools in the soil organic module of the DAISY model is original and very interesting, and their draft is well structured and written. However, I have a major concern regarding the rationale of the DRIFTS stability index of SOM in mineral soil, and its use for partitioning the C kinetic pools of SOM dynamics models. In this review, I will only discuss this concern, though this stimulating and timely work would deserve many other comments, as highlighted by the two other reviewers of this draft. First, I would like to come back on the justification of the DRIFTS stability index by Demyan and colleagues in their 2012 paper. Demyan et al. (2012) searched for information related to SOM in DRIFTS spectra of bulk mineral soils, and its link to SOM stability as assessed by a SOM density fractionation scheme. In their search for SOM information in DRIFTS spectra of bulk mineral soils, they discarded "wavenumbers of functional groups associated with non-organic compounds such as silicates and alumino-iron oxides". For them, "these criteria removed the peaks <1000 cm−1 and the peaks at 1980, 1870, 1792 and 1390 cm−1", but not the 1620 cm−1 peak. For them, "the [DRIFTS] peak at 1620 cm−1 was assigned to predominately aromatic C =C stretching and/or asymmetric–COO−stretching but possibly also C = O vibrations". Demyan et al. (2012) show that "a positive relationship was found between the ratio of the peaks at 1620 and 2930 cm−1 (1620:2930) and the ratio of stable C (sum of C contained in clay and >1.8 g cm−3 fractions) to labile C (amount of C in the <1.8g cm−3 fraction) (R2= 0.62, P = 0.012)." For the authors, this result justifies that the DRIFTS stability index can reliably be "taken as an indicator of SOM stability" (Demyan et al., 2012).

We originally stated (line 369ff) that the peaks were selected in order to have limited mineral interference (e.g. Demyan et al., 2012). In their original publication only soils with the same mineral background were taken as additional measure of caution. As this approach showed potential for Bad Lauchstädt, we thought that this could justify trying to use DRIFTS as a general stability index. This was the reason for conducting this study. We are aware of the mineral signal at the 1620 cm$^{-1}$ region and this fact was also acknowledged in the original publication of Demyan et al. (2012). By carefully selecting the integration limits, it should be possible to minimize the mineral interference and get a general applicable stability index. We aimed to combine several sites to have several test cases. The reason for the statistical analysis of the model error was exactly that we wanted to test whether the DSI is a useful proxy across a range of sites. We state some further reasoning below why we think the 1620 cm$^{-1}$ peak as we selected is representative of aromatic carbon and what was changed in the main text.

However, a short look at the literature on DRIFTS of soils show that the 1620 cm-1 peak in bulk mineral soils cannot be exclusively assigned to absorption from SOM functional groups (C = C or C = O) as claimed by Demyan et al. 2012. I will only cite two important papers: Nguyen et al. (1991) and Reeves (2012).Nguyen and colleagues, based on DRIFTS spectra of pure mineral compounds and various soil samples demonstrated that "The DRIFT spectra of soils containing organic matter show considerable overlap of the silicate combination bands in the 2000-1600 cm-1 region". I provide here the Figure 1 modified from Nguyen et al. (1991) showing the DRIFTS spectra of quartz (pure or diluted in KBr), highlighting the strong absorption of quartz at 1620 cm-1 (for the DRIFT spectra of pure quartz). They suggested that "Spectral subtraction techniques or prior chemical treatment may thus be required to resolve these peaks." (Nguyen et al., 1991).

Reeves (2012) based on works similar than Nguyen et al. (1991), concluded that "With the exception of the bands at 2930and 2850 cm−1 due to aliphatic CH [when the soil does not contain carbonates, added by me] and the large OH band spanning most of the region between 2700 and 3500cm−1, there is little that is obviously due to OM in the soil spectra". Regarding the 1620cm-1 DRIFTS peak, he suggested, following Nguyen et al. (1991) that "the region between 1750–1600 cm−1 can be interpreted, despite the presence of strong silica bands, because silica can be ash subtracted quite well". But he also concluded his paper with this warning regarding spectral subtraction: "It will detect not only whether your sample is changed by 0.1% at some point in time, but will also seem to detect the phases of the moon and the mood you were in while you were measuring the data."(Hirschfeld, 1984; cited by Reeves, 2012).I deduce from this short literature survey that in their 2012 paper, Demyan et al. incorrectly assigned to SOM compounds (C = C, C = O) exclusively the 1620 cm-1 DRIFTS peak of bulk mineral soils, as this peak is also due to mineral compounds such as quartz (but also to water in some phyllosilicates).

It is not correct that we claimed an "exclusive" assignment of the 1620 cm$^{-1}$ peak to SOM functional groups, but rather that by carefully selected integration limits, the delimited area of the 1620 cm$^{-1}$ is mostly representative of those organic groups.

In fact, the different spectra of soils before and after ashing or pyrolysis (as the example below taken from the supplementary material of Nkwain et al. (2018)) demonstrate that a considerable part of the delimited 1620 cm$^{-1}$ peak is lost. Demyan et al. (2013) found a decrease in absorbance intensity at 1620 cm$^{-1}$ with maximum losses occurring between 400-500°C (Figure S8, Left) for bulk soils. In the same study separated fractions were also analyzed, with a similar 1620 cm$^{-1}$ peak loss found for particulate organic matter (POM) that was assumed to be mineral free. These consistent findings of the organic contribution to the 1620 cm$^{-1}$ peak from both rapid pyrolysis and in situ thermal monitoring of soil samples up to 700 °C where also found when pretreating bulk soil or fractions with NaOCl (Yeasmin et al., 2017).

[Figure]

**Figure S7. DRIFTS spectra of (a) unpyrolyzed soil and (b) pyrolyzed soil from Bad Lauchstädt (FYM). From (Nkwain et al., 2018)**

[Figure]

**Figure S8 Left: Change of C-H (2930 cm−1 ) and (b) C = O/C = C (1620 cm−1 ) vibrations with heating as measured by In situT DRIFTS of bulk soil samples from Bad Lauchstadt, ¨ Kraichgau, and Swabian Alb. Right: Change of C-H (2930 cm−1 ) and (b) C = O/C = C (1620 cm−1 ) vibrations measured in bulk soil and fractions of soils from Kraichgau and Swabian Alb (Demyan et al., 2013). *POM-particulate organic matter, Sa+A-sand and stable aggregates, Si+C-silt and clay, rSOC-resistant soil organic carbon.**

We would like to draw the attention to the way we selected the integration limits. We only take the top of the larger 1620 cm⁻¹ peak. As the three examples above demonstrate, this is mostly the part

which is removed by burning, pyrolyzing or NaOCl treatment. See the picture below for typical peak areas from our samples.

[Figure]

However, as we recently demonstrated (Laub et al., 2019), and further found in the current study, the 2930 cm$^{-1}$ peak is also subject to interference even in non-carbonate containing soils. This is mostly by water, which can partly be removed by higher drying temperatures.

To further illustrate how the 1620 cm-1 DRIFTS peak of bulk mineral soil is poorly related to SOM compounds, I provide the Figure 2 based on published and unpublished data from the paper of Barré et al. (2016) in Biogeochemistry showing the non-parametric Spearman's Rho coefficient of DRIFTS spectra from soils coming from the Ultuna Fame trial, one site that was used in this reviewed work by Laub and colleagues, with SOC concentration. In Figure 2, we clearly see the strong and positive Rho coefficient of the 2900 cm-1 spectral region with SOC concentration while the 1620 cm-1spectral region show a Rho coefficient with SOC concentration close to 0, suggesting (though not demonstrating) that other compounds that organic matter absorb energy in the 1620 cm-1 spectral region of DRIFTS spectra, when scanning bulk mineral soils. From the above-mentioned information, I therefore question the rationale of the DRIFTS stability index of soil organic matter (SOM) in mineral soil samples.

Actually, we found that the 1620 cm$^{-1}$ was mostly negatively correlated with TOC, but in our recent publication, we showed that there was a slight positive correlation with TOC, if the 1620 cm$^{-1}$ peak was normalized (divided by the 1880 cm$^{-1}$ quartz/ silicates Peak) (though not significant for the small number of 21 archive samples we used within this study. See the supplementary material of Laub et al., (2019).

| correlation of | TOC | TOC | TOC | TOC | TOC | TOC | TOC | TOC |
| with | 2930 cm$^{-1}$ | n2930 cm$^{-1}$ | 1620 cm$^{-1}$ | n1620 cm$^{-1}$ | 1530 cm$^{-1}$ | n1530 cm$^{-1}$ | 1159 cm$^{-1}$ | n1159 cm$^{-1}$ |
| temperature | | | | | | | | |
| 32 | 0.77 | 0.85 | -0.69 | 0.14 | -0.70 | -0.85 | -0.60 | 0.18 |
| 55 | 0.73 | 0.77 | -0.64 | -0.01 | -0.66 | -0.83 | -0.56 | 0.08 |
| 65 | 0.75 | 0.81 | -0.63 | 0.06 | -0.68 | -0.84 | -0.55 | 0.11 |
| 75 | 0.76 | 0.82 | -0.62 | 0.12 | -0.68 | -0.80 | -0.56 | 0.18 |
| 85 | 0.74 | 0.79 | -0.63 | 0.09 | -0.67 | -0.81 | -0.56 | 0.15 |
| 95 | 0.72 | 0.80 | -0.63 | 0.19 | -0.68 | -0.84 | -0.57 | 0.24 |
| 105 | 0.76 | 0.83 | -0.62 | 0.16 | -0.68 | -0.81 | -0.58 | 0.20 |

My interpretation is that this index is dividing a quantity that is highly correlated to SOC concentration (the 2900 cm-1 spectral region), by a quantity that is weakly changing when SOC concentration is modified (the 1620 cm-1 spectral region, provided a similar mineral composition). The DRIFTS stability index may thus show an increased SOC lability when SOC concentration is increased. I thus hypothesize that the DRIFTS stability index, as proposed by Demyan et al. (2012) and Laub and colleagues in this reviewed draft, may provide some information that is basically the same (though with added noise) than a variable much simpler than their index: total SOC concentration.

We agree with the interpretation that the DSI is "dividing a quantity that is highly correlated to SOC concentration by a quantity that is weakly changing when SOC concentration is modified", and as we demonstrate above, both quantities are linked to forms of SOC. That the 1620 cm$^{-1}$ does not change strongly with SOC content, while, as destructive techniques demonstrate, it is still consisting mostly of aromatic carbon compounds (according to our integration limits), is exactly the reason why it is a very suitable proxy for slow turnover SOC.

It is well documented that an increase in SOC concentration is associated with an increased in the labile/stable SOC ratio, and all proposed indicators of SOM stability should be compared to SOC concentration, the most simple and straightforward indicator of SOM stability (though not very accurate).

What is the Spearman's Rho coefficient of the DRIFTS stability index with SOC concentration in the dataset of Laub and colleagues?

We calculated the Pearson`s correlation coefficient -0.57 and Spearman`s rank correlation coefficient to be -0.68 between OC content and the DSI (as in formula 2) for the whole dataset (n=50). See the plot below

[Figure]

We think, that the nonlinearity of the relationship between the DSI and the SOC content points towards the possibility, that as SOC increases, most of the carbon is added to the fast turnover pool and that this could potentially be lost rather fast again.

I suggest that the authors (rather than using the spectral subtraction technique suggested by Nguyen et al., 1991 or Reeves, 2012), (i) test a soil dilution in KBr to reduce mineral artifacts in the 1620 cm-1 spectral region of neat DRIFTS, (ii) or test attenuated total reflectance mid-infrared spectroscopy (MIR-ATR) as an alternative technique.

Dilution with KBr (1:3 and 1:100) had been tested by Demyan et al. (2012), mainly to determine whether there was specular reflectance in the 1159 cm$^{-1}$ region, but was not found to yield better performances for deriving the DSI. Using neat samples avoids hygroscopic KBr which would have the potential to absorb water interfering with the 2930 cm$^{-1}$ peak and other non-desired interactions with the sample. We think that the major advantage of the DSI and DRIFTS PLSR is that it is possible to use undiluted bulk soil samples, and that it is nondestructive (cost effective and other analysis can be done with the same samples). We see the major advantage also in large scale applications, such as regional simulations, where other techniques are either too expensive or time consuming.

Indeed, MIR-ATR is a technique where the 1620 cm-1 peak region seems to be much less affected by quartz and other minerals that neat DRIFT signal, as illustrated in Figure 3 (Cécillon, Unpublished data).

From our understanding, the issue with ATR usually is that the signal throughput to the detector is weaker, thus the overall spectral features stand out less and are dominated by the silica vibrations at <1500 cm$^{-1}$, which is also shown in Figure 3 of Lauric Cécillon´s comment. The maximum absorbance in the DSI wavenumbers is almost an order of magnitude lower as compared to DRIFTS. If you zoom in on the figure, you can also see a small peak probably around 1620 cm$^{-1}$ in the silica sample, so it seems not to be free of mineral interference.

It might be possible that MIR-ATR is an alternative to DRIFTS, if it can reduce mineral interference at the 1620 cm$^{-1}$, but given the less strong signal of organics peaks it might be limited in low C soils. It could be worthwhile to do further research towards that direction and we think that this could be the content of another future publication.

Finally, as Laub and colleagues benefit from soil samples from two long-term bare fallow sites in Europe, I suggest that they compute the Spearman's Rho coefficient of their DRIFTS stability index

with the proportion of centennially persistent soil organic carbon(CPsoc), that may be derived from the SOC evolution in the bare fallow plots, as shown by Cécillon et al. (2018).

A higher Spearman's rho coefficient of the DRIFTS stability index with CPsoc than the Spearman's rho coefficient of SOC concentration with CP-soc, would suggest an added value of the index compared to SOC concentration, in its current state.

We cannot follow the reasoning here. We computed CPsoc with the value of 6.95 g kg$^{-1}$ from Ultuna derived by Cécillon et al. (2018). As shown below, by definition of CPsoc, Spearman`s correlation coefficient (or in our graph an $R^2$ of an exponential equation, which is effectively the same) between SOC% and CPsoc of SOC is perfect (1!) if we only use data from one long term site. To resolve this, one would need data for the CPsoc and DSI from many long term experiments.

[Figure]

Additionally, the reasoning behind CPsoc is only valid for RothC type models, which assume that there is only one actively decomposing SOC pool and another passive SOC pool which is NOT subject to decomposition. In this study, we worked with DAISY, which is a CENTURY type model, that has a fast and slow SOC pool, both subjected to decomposition. We think that this is in agreement with the principle behind DRIFTS, and that microorganisms primarily target high energy aliphatic SOC, but aromatic SOC is also decomposed at a much slower rate.

Overall, we very much acknowledge the issue of mineral interference addressed by the reviewer (see line 369 in the original manuscript) and the new addition:

> Margenot et al., 2015). While both peaks are subject to interference (2930 cm$^{-1}$ from water and 1620 cm$^{-1}$ from minerals mainly (Nguyen et al., 1991), it should be possible to limit the interference with carefully selected
>
> integration limits. Indeed, Demyan et al. (2012) found aliphatics to be enriched under long-term farmyard manure application and depleted in mineral fertilizer or control treatments, and showed that the ratio of the 2930 cm$^{-1}$ to 1620 cm$^{-1}$ peaks had a significant positive correlation with the ratio of labile to stable SOM obtained by size and density fractionation. It was further corroborated that the part of the peaks they used mainly select the top part of the peak areas, which are lost during combustion (Demyan et al., 2013). Hence, we hypothesised that

We have addressed this issue mainly by carefully delaminating the integration area and now have more clearly pointed to this in the methods:

the aliphatic C-H streching, 1660 – 1580 cm$^{-1}$ for aromatic C=C streching vibrations). The carefully selected integration limits, as well as using only the peak area on top of the local baselines were critical to reducing signal inteference from water and minerals. Additionally, soils from the experiments in Kraichgau and Swabian Jura

We have further added a more detailed discussion on open questions of the DSI in the new manuscript version and finalize the section with the limitation that DSI should be tested before used with different soil types.

There are some remaining questions that should be answered to standardize the application of the DSI for model initialization. Those are related to how the type of spectrometer influences the spectra, as well as how water and mineral interferences (Nguyen et al., 1991) in the spectra can be eliminated or at least be further reduced. We had the experience, that spectra and therefore peak areas vary to some degree between different spectrometers (mostly due to different detectors, types of detector cooling and resolution). Hence, it will be necessary to either use the same spectrometer, or to develop techniques to standardize spectra across a large number of instruments. While a careful selection of integration limits and the use of a local baseline minimizes mineral interference of DRIFS spectra from bulk soils, and mostly selects the part of the 1620 cm$^{-1}$ peak which gets usually lost when SOC is destroyed (Demyan et al., 2013; Yeasmin et al., 2017), it is not possible to completely eliminate mineral interference with DRIFTS. The recent coupling of pyrolysis with DRIFTS (Nkwain et al., 2018) might be a further advancement of the DSI, as it overcomes mineral interferences in the spectra. However, this technique is more complex due to a larger number of visible organic peaks, including CO$_2$ that develops from the pyrolysis, which makes it not easily applicable to established two-pool models such as DAISY. In addition, a considerable portion (30 – 40 %) of SOM is not pyrolyzed and therefore not recorded in the spectra. So iIn summary, even despite of the acknowledged shortcomings, it was found that the DSI can be directly used to distribute SOM between pools in two pool models, though there is some mineral interference. Furthermore, DSI was suitable for a wide range of soils and improved model performance. Hence, DSI seems to be a more robust proxy for pool initialization then methods such as steady state or long-term spin-up runs which rely on strong assumptions to which they are very sensitive though there is very limited data to prove them. It would still be advisable to test the DSI before using it for mineralogy that differs considerably from the soils of this study.

In the quest to find measurable fractions for model pools, we think that the DSI is a useful proxy (carefully selected integration limits, nonlinear relation with SOC, evidence that our 1620 cm$^{-1}$ is mostly from carbon, drying at 105 °C to reduce water interference at 2930 cm$^{-1}$). We first and foremost consider the DSI as a potential proxy to help initializing two pool SOM models, and our question was, whether it was useful for this purpose or not, compared to steady state initializations. We think the value of this publication is to establish that the DSI has the potential to be a measurable fraction as a model pool proxy and thereby reducing model uncertainty, and show this to the scientific community. As any research this opens new questions which could lead to further development and refinement of the DSI. We think, that our study could demonstrate the DSI's usefulness and that it might be worthwhile to put further efforts and research towards its validation, use or optimization, especially because of its ease of use and inexpensive nature.

References:

Demyan, M. S., Rasche, F., Schulz, E., Breulmann, M., Müller, T. and Cadisch, G.: Use of specific peaks obtained by diffuse reflectance Fourier transform mid-infrared spectroscopy to study the composition of organic matter in a Haplic Chernozem, Eur. J. Soil Sci., 63(2), 189–199, doi:10.1111/j.1365-2389.2011.01420.x, 2012.

Laub, M., Blagodatsky, S., Nkwain, Y. F. and Cadisch, G.: Soil sample drying temperature affects specific organic mid-DRIFTS peaks and quality indices, Geoderma, 355, 113897, doi:10.1016/j.geoderma.2019.113897, 2019.

Nkwain, F. N., Demyan, M. S., Rasche, F., Dignac, M.-F., Schulz, E., Kätterer, T., Müller, T. and Cadisch, G.: Coupling pyrolysis with mid-infrared spectroscopy (Py-MIRS) to fingerprint soil organic matter bulk chemistry, J. Anal. Appl. Pyrolysis, 133(April 2017), 176–184, doi:10.1016/j.jaap.2018.04.004, 2018.

Yeasmin, S., Singh, B., Johnston, C. T. and Sparks, D. L.: Evaluation of pre-treatment procedures for improved interpretation of mid infrared spectra of soil organic matter, Geoderma, 304, 83–92, doi:10.1016/j.geoderma.2016.04.008, 2017.

---

## Author Response (AR1)

Dear Michael Weintraub,

We are pleased to upload our revised manuscript in response to the comments of the three reviewers. We have made a number of changes in the manuscript in order to address the concerns of the reviewers, which we found very constructive, objective and of high quality. In our initial response letters, we have included detailed answers to each concern including what we changed in the manuscript as a result. Therefore, we will summarize here the most important changes from our point of view.

As we understood it, the main concerns of Sander Bruun were, 1.1) whether the DRIFTS stability index (DSI) changes with type of mid-infrared spectrometer bench used (in our experience it slightly does), 1.2) why we used a statistical analysis of model error (mainly to interpret a temporal trend), 1.3) whether this Bayesian model calibration could be data limited (we rather think that it is due to model structure) and 1.4) if we could include graphs of the DSI change over time of the long-term experiments (we did so). To all these points we made changes in the text and added additional figures where necessary to further elaborate our point.

Additionally to some concerns shared with Sander Bruun, the main concerns of reviewer 2 were, 2.1) whether the DSI was a proxy supporting the hypothesis of resistance due to molecular properties of SOC only (to our understanding the DSI is also related to other forms of stabilization), and 2.2) whether the DSI might be also useful for models using microbial mineral models (to our understanding yes, as it correlates also well with size density fractionation) and 2.3) that we change the wording of some claims in the abstract, so that they are not misleading. To each of these concerns we made changes in the text to address them. The suggested spell and language checking was done.

As we understood it, the main concerns of Lauric Cécillon were: 3.1) whether the mineral interference of the 1620 cm$^{-1}$ peak would mean that we only divide the 2930 cm$^{-1}$ peak by some "artifact". By carefully selecting integration limits, hence the part of the 1620 cm$^{-1}$ peak we use relates mostly to carbon. 3.2) Whether the DSI was more informative than the SOC content alone (the strong correlation to % of centennially persistent SOC across sites seems to indicate so) and 3.3) that we should discuss the shortcomings and the "known unknowns" of the DSI, due to mineral interference and other factors in more detail. We followed the advice by adding an additional figure (Figure S1 in updated manuscript – correlations with CPsoc), as well as adding a thorough discussion of the mineral, water and spectrometer interference that the DSI is subject to. We agree with you, that these concerns are likely shared by other readers and found the discussion with Lauric to be especially fruitful.

We thank you for handling the manuscript and are happy that it is considered to be of interest to the readers of Biogeosciences.

With kind regards on behalf of all coauthors,

Moritz Laub

**Responses to Reviewers:**

We would like to thank the editor for taking the time to handle our manuscript and for finding three very constructive reviewers. We also want to thank all reviewers for taking the time and reviewing our manuscript to help improve its quality. We are grateful for the honest and thorough feedback. The suggestions were highly useful and provided us with information, where misunderstandings could be possible and where we needed to make our message clearer and to discuss the limitations of the DSI in more detail. They helped to further improve the quality of this manuscript and we hope that we addressed concerns to a satisfying extent. Our comments to the reviewers in the following are in blue color. We made use of the constructive criticism and altered the text of the manuscript, where applicable. We added screenshots of alterations in the text related to the comments. These are displayed in green color.

**1.1   Reviewer 1: Sander Bruun**

**General comments**

The papers deals with initialization of pools in the soil organic matter model of Daisy. The paper is using unique datasets for long-term fallow treatments to test a new way of initializing the soil organic matter pools based on specific peaks in the DRIFTS specta of the soils. Pool initialization of SOM model is an important issue that is still causing some difficulties with the currently used approaches. The paper is therefore very timely and present an interesting approach that could be useful in many situations. The work is of a high quality and based on high quality data and the manuscripts is well written.

**Specific comments**

Line 78. I agree that the DSI can be better than the steady-state assumption, but perhaps it is worth discussing this in a little more detail. If information about the history of the site is available then that method should work. This require that the history is known for millennia, and that is rarely the case.

We added one more sentence as suggested, at line 56.

Line 118: Was soil samples from throughout the experimental period analyzed? Please specify.

Yes, from throughout the period. We specified this more.

Line 129: The spectra were not recorded in absorbance, but subsequently converted to absorbance units, right?

Yes – wording changed

Line 130: I wonder how much this way of determining the DSI is affected by the instrument i.e. if somebody took the same soils and did the measurement on another instrument would the get the same DSI and pool sizes. I am afraid that it would be quite much affected by that especially if you use other IR detection techniques. Maybe it would be worth addressing this in the discussion.

Indeed, at least to our experience, there are some differences between the spectra of different spectrometers. We added a sentence addressing this in chapter 4.1. As we already tested different temperatures for drying, which we found to be the most dominant factor affecting DSI, it was beyond the scope of this publication to test the effect of the Spectrometer. We were first and foremost interested in, whether the DSI approach adds value in general to SOM initialization, which we think it does.

Line 181: It says 84% and not 83% in Table 2. Please correct where appropriate.

Done

Line 196 to 209: I am not entirely sure I understand what the point of analyzing the SMEx with a statistical model is. I think you should consider whether it add enough understanding to warrant inclusion. Alternatively explain the point a little better.

We wanted a more sophisticated analysis of the model error than and since in some experiments (Swabian Jura and Kraichgau) we had several fields, make use of the statistical power provided by the experimental design. The second advantage of a statistical analysis of model error was, that we could analyze for a time trend (increase with time) of the model error. Obviously, the results still only hold for the fields we analyzed.

Line 236-237. The necessity of constraints on the fSOM-Slow parameter is a little problematic. I cannot help thinking that it means that the data, which is used for calibration, is insufficient. With these restraints, I guess you are likely to end up with a value of 0.35 which is rather arbitrarily chosen by you.

We fully agree with this statement. Actually, we also found containing a bit problematic, but a humification of 95% or more also did also not seem realistic. For us, this is an indicator how model structure affect the results. This is why we came up with a possible alternative formulation of DAISY. Actually, we tested the new structure of DAISY and found, that with this fSOM_s does not run into the constraint anymore, even if we do not constrain it. See as an example the results of the new structure with (2) and without (3) the fSOM_s constraints compared to (1) the initial BC of this study:

[Figure]

Line 364-365. I agree that even though we have had the same management for a longtime the steady-state assumption is not valid. However, I believe that the reason for this has to do with

longer-term effects rather than the smaller effects that you mention i.e. variation in climate agricultural management. If you look at a longer terms, most sites would probably have been deforested within the last 2000 years. Because of the high inputs from the forest, this could have resulted in an unusually large fraction of resistant organic matter that has not been degraded from that period. Also it is very common with drained soils soil. This means that the soil at some time it its history has had a very high water table and perhaps even been inundated. We know that this can result in significant accumulation of organic matter. After the soil has been drained, this has led to a large residual of resistant C again. The same could happen if there has been a history of fires with inputs of charcoal. Perhaps this is worth discussing a bit more.

We agree and added these possibilities to the main text.

Line 373: I cannot help it thinking that it is somewhat of a coincidence that you get better model performance with the DSI as long as you have not recalibrated the model. Of course using more data as for example DSI to restrain the model should improve the model, but only after it has been recalibrated.

We interpreted the fact that SMB-C simulations were best when using the DSI as indicator that it is a proxy of generally utility, even if the turnover rates are unclear. As SMB-C is a much faster reacting pool than TOC, which did not change that much in our trials in Kraichgau and Swabian Jura. The DSI at 105°C was consistently lower in model error for simulated SMB-C than the steady state initialization, which we saw as an indicator that it is a useful proxy regardless of turnover rate, as long as there is a clear distinction between fast and slow pools.

It is not entirely clear what data were used for the calibrations based on DSI. As far as I understand, you measured DSI of all the soil samples and that means that you can compare the simulated distribution between fast_SOM1 and slow_SOM with the one measured and calculated using formula (2) and a similar formula for fast_SOM. Is this the case? And if it is why have you not shown the "measured" value of fast and slow SOM and compared it with the modelled?

You are correct, that we used the measured DSI throughout the simulation period for the Bayesian calibration. We are happy to provide the modelled vs measured DSI throughout the simulation period – we also added it to the manuscript as additional figure s7:

[Figure]

**Figure S 1 Development of simulated vs observed SOM in the slow pool, according to DSI division throughout the simulation period (for brevity only for 105 °C). Bars indicate standard deviation of all plots per field.**

Is it worth publishing the optimal parameters selected by the Baysian calibration based on DSI?

While we think that the ideal way to use our results is using the posterior probability distributions of our parameters, we have mentioned the parameter set of the maximum likelihood from our

Bayesian calibration in chapter 3.3 (0.34, 2.29 * $10^{-4}$, 3.25 * $10^{-5}$ for the original weight calibration and 0.06, 9.58 * $10^{-5}$ and 5.54 * $10^{-5}$ for the calibration using original weights and no DSI).

**1.2 Reviewer 2:**

**General comments**

Laub and Colleagues present interesting ideas how DRIFTS spectra could be used to initialize and calibrate soil organic matter models. What warrants more discussion is that with their results we should put again more emphasis on the chemical recalcitrance hypothesis, i.e. that molecular properties determine the persistence of organic matter in soils. The literature seems to disagree (Schmidt et al., 2011). If we indeed assign the aromatic peak to slow cycling pools with a turnover time of 426 years and the aliphatic peak to a fast cycling pool with 47 to 90 years, the authors would contradict the synthesis of Schmidt et al. (2011) (their Figure 1, for example).

We do not think, that it contradicts Schmidt et al. (2011). Rather the DSI seems to point towards the same direction as other measures of SOM quality, such as the amount of SOM in different aggregate sizes and fractions. This was actually shown in our previous works (Demyan et al., 2012). We have to keep in mind that the DSI is still only a proxy and dividing the whole continuum of SOM quality into two "qualities" is a strong simplification of the real world. However, we think it seems to be a valid one, especially when two pools SOM models are to be used, which anyway divide SOM into two pools.

In my opinion, it would be interesting if the authors could at least discuss how their DRIFTS peaks could be useful for the new class of microbial-mineral models such as Tang and Riley (2015) or Sulman et al., (2014)

We think that DRIFTS could also be useful for those models, because of a good correlation of the DSI to size density fractionation (Demyan et al., 2012), which is thought more representative of structural protection mechanisms. We added one sentence about this in the discussion. As Tang and Riley (2015) stated, it is not likely that CUE and Q10 are static (or the same for different complexities of SOM), which to our opinion points to the need of reliable pool partitioning proxies. As we already addressed thin issue in the manuscript (line 528ff), we did not add anything new.

**Specific comments**

The authors state that "the DRIFTS initialization of SOM pools significantly reduced model errors of poor performing model runs assuming steady state, irrespective of the turnover rates used, but the faster turnover parameter set fit better to all sites except Bad Lauchstädt. This suggests that soils under long-term agricultural use were not necessarily at steady state." In my opinion this statement is not backed up by their results. The Bruun parameters with steady state assumption perform better at Ultuna and Kraichgau + Swabian Jura (Table 4) for SOC stocks.

We agree that it was a bit oversimplified, so we altered the wording. For this statement we placed more weight on the Kraichgau + Swabian Jura sites than Ultuna, because those consisted of six fields. What we saw there was a significant improvement of the sensitive SMB-C, for both turnover rates, while for SOC stocks, there were only significant differences in model error between turnover rates but not between initializations.

The authors also state that "[...] two approaches [...] significantly reduced parameter uncertainty and equifinality". One of the approaches was the inclusion of DRIFTS. But looking at the violin plots in Figure 5, only the humification efficiency seems to be better constrained. I suggest modifying the statement towards this direction.

It is true, that humification efficiency was the parameter most seriously constrained by the DSI, also the turnover of the slow carbon pool was stronger constrained (standard deviation of $9.3 * 10^{-6}$ with DSI vs $12.3 * 10^{-6}$ without DSI). We altered the wording in the sentence, to be more accurate.

I agree with the other reviewer, Sander Bruun, that analyzing the squared model errors with a statistical model should at least be better explained.

This was done, see the comment to Sander Bruun, above.

The manuscript would benefit from a thorough spell and language check.

This will be done on the final reviewed version manuscript.

**Comments of Reviewer 3: Lauric Cécillon**

"Reservation on the rationale of the DRIFTS stability index of soil organic matter (SOM) in mineral soil, and its use for partitioning the C kinetic pools of SOM dynamics models" This draft by Laub and colleagues describes a method to divide soil organic matter(SOM) into fast and slow cycling C pools in the soil organic module of the DAISY model. This method is based on the characterization of bulk mineral soil samples using mid-infrared diffuse reflectance spectroscopy (DRIFTS). DRIFTS spectra of bulk mineral soils are used to compute the "DRIFTS stability index" of SOM, defined as the ratio of aliphatic C-H (2930 cm-1) to aromatic C=C (1620 cm-1) stretching vibrations.

The DRIFTS stability index was previously published by Demyan et al. (2012) in the European Journal of Soil Science.

The development of routine and operational method to initialize the relative size of C kinetic pools from SOM dynamics models is a very important and timely topic. Indeed, the accuracy of the simulations of SOM evolution in mineral soils by current models is strongly questioned, notably because a poor initialization of the size of C kinetic pools. The method proposed by Laub and colleagues, using the DRIFTS stability index to divide soil organic matter (SOM) into fast and slow cycling C pools in the soil organic module of the DAISY model is original and very interesting, and their draft is well structured and written. However, I have a major concern regarding the rationale of the DRIFTS stability index of SOM in mineral soil, and its use for partitioning the C kinetic pools of SOM dynamics models. In this review, I will only discuss this concern, though this stimulating and timely work would deserve many other comments, as highlighted by the two other reviewers of this draft. First, I would like to come back on the justification of the DRIFTS stability index by Demyan and colleagues in their 2012 paper. Demyan et al. (2012) searched for information related to SOM in DRIFTS spectra of bulk mineral soils, and its link to SOM stability as assessed by a SOM density fractionation scheme. In their search for SOM information in DRIFTS spectra of bulk mineral soils, they discarded "wavenumbers of functional groups associated with non-organic compounds such as silicates and alumino-iron oxides". For them, "these criteria removed the peaks <1000 cm−1 and the peaks at 1980, 1870, 1792 and 1390 cm−1", but not the 1620 cm−1 peak. For them, "the [DRIFTS] peak at 1620 cm−1 was assigned to predominately aromatic C =C stretching and/or asymmetric–COO−stretching but possibly also C = O vibrations". Demyan et al. (2012) show that "a positive relationship was found between the ratio of the peaks at 1620 and 2930 cm−1 (1620:2930) and the ratio of stable C (sum of C contained in clay and >1.8 g cm−3 fractions) to labile C (amount of C in the <1.8g cm−3 fraction) (R2= 0.62, P = 0.012)." For the authors, this result justifies that the DRIFTS stability index can reliably be "taken as an indicator of SOM stability" (Demyan et al., 2012).

> We originally stated (line 369ff) that the peaks were selected in order to have limited mineral interference (e.g. Demyan et al., 2012). In their original publication only soils from the same field experiment with the same texture and mineral background were taken as additional measure of caution. As this approach showed potential for the site at Bad Lauchstädt, we hypothesized that this could justify evaluating the use of the DRIFTS 1620:2930 ratio as a more general stability index. We are aware of the mineral signal in the vicinity of the 1620 cm$^{-1}$ peak and this fact was also acknowledged in the original publication of Demyan et al. (2012). By carefully selecting the integration limits, it was possible to minimize the mineral interference and get a general applicable stability index (see evidence below). In the current study, we aimed to combine several sites with differing textures and mineralogies to have several test cases. The reason for the statistical analysis of the model error was exactly that we wanted to test whether the DSI is a useful proxy across a range of sites. We state some further reasoning below why we think the

1620 cm⁻¹ peak and the specific peak limits that we have used (1660 – 1580 cm⁻¹) is representative of aromatic carbon and what was changed in the main text.

However, a short look at the literature on DRIFTS of soils show that the 1620 cm-1 peak in bulk mineral soils cannot be exclusively assigned to absorption from SOM functional groups (C = C or C = O) as claimed by Demyan et al. 2012. I will only cite two important papers: Nguyen et al. (1991) and Reeves (2012).Nguyen and colleagues, based on DRIFTS spectra of pure mineral compounds and various soil samples demonstrated that "The DRIFT spectra of soils containing organic matter show considerable overlap of the silicate combination bands in the 2000-1600 cm-1 region". I provide here the Figure 1 modified from Nguyen et al. (1991) showing the DRIFTS spectra of quartz (pure or diluted in KBr), highlighting the strong absorption of quartz at 1620 cm-1 (for the DRIFT spectra of pure quartz). They suggested that "Spectral subtraction techniques or prior chemical treatment may thus be required to resolve these peaks." (Nguyen et al., 1991).

Reeves (2012) based on works similar than Nguyen et al. (1991), concluded that "With the exception of the bands at 2930and 2850 cm−1 due to aliphatic CH [when the soil does not contain carbonates, added by me] and the large OH band spanning most of the region between 2700 and 3500cm−1, there is little that is obviously due to OM in the soil spectra". Regarding the 1620cm-1 DRIFTS peak, he suggested, following Nguyen et al. (1991) that "the region between 1750–1600 cm−1 can be interpreted, despite the presence of strong silica bands, because silica can be ash subtracted quite well". But he also concluded his paper with this warning regarding spectral subtraction: "It will detect not only whether your sample is changed by 0.1% at some point in time, but will also seem to detect the phases of the moon and the mood you were in while you were measuring the data."(Hirschfeld, 1984; cited by Reeves, 2012).I deduce from this short literature survey that in their 2012 paper, Demyan et al. incorrectly assigned to SOM compounds (C = C, C = O) exclusively the 1620 cm-1 DRIFTS peak of bulk mineral soils, as this peak is also due to mineral compounds such as quartz (but also to water in some phyllosilicates).

It is not correct that we claimed an "exclusive" assignment of the 1620 cm⁻¹ peak to SOM functional groups, but rather that by carefully selected integration limits, the delimited area of the 1620 cm⁻¹ is mostly representative of those organic groups.

In fact, the different spectra of soils before and after ashing or pyrolysis (as the example below taken from the supplementary material of Nkwain et al. (2018)) demonstrate that a considerable part of the delimited 1620 cm⁻¹ peak is lost. Demyan et al. (2013) found a decrease in absorbance intensity at 1620 cm⁻¹ with maximum losses occurring between 400-500°C (Figure S8, Left) for bulk soils. In the same study separated fractions were also analyzed, with a similar 1620 cm⁻¹ peak loss found for particulate organic matter (POM) that was assumed to be mineral free. These consistent findings of the organic contribution to the 1620 cm⁻¹ peak from both rapid pyrolysis and in situ thermal monitoring of soil samples up to 700 °C where also found when pretreating bulk soil or fractions with NaOCl (Yeasmin et al., 2017).

[Figure]

**Figure S7. DRIFTS spectra of (a) unpyrolyzed soil and (b) pyrolyzed soil from Bad Lauchstädt (FYM). From (Nkwain et al., 2018)**

[Figure]

**Figure S8 Left: (a) Change of C-H (2930 cm$^{-1}$ ) and (b) C = O/C = C (1620 cm$^{-1}$ ) vibrations with heating as measured by in situT DRIFTS of bulk soil samples from Bad Lauchstadt, ¨ Kraichgau, and Swabian Alb. Right: (a) Change of C-H (2930 cm$^{-1}$ ) and (b) C = O/C = C (1620 cm$^{-1}$ ) vibrations measured in bulk soil and fractions of soils from Kraichgau and Swabian Alb (Demyan et al., 2013). *POM-particulate organic matter, Sa+A-sand and stable aggregates, Si+C-silt and clay, rSOC-resistant soil organic carbon.**

We would like to draw the attention to the fact that by a careful selection of the integration limits, we only take the top of the larger 1620 cm$^{-1}$ peak (which in our samples made up 15 to 33% of the whole peak area). As the three examples above demonstrate, this is mostly the part,

which is removed by burning, pyrolyzing or NaOCl treatment. This is the same principle as used for the aliphatic peak area at 2930 cm$^{-1}$, which is on top of a larger OH peak and to our knowledge, there is little debate about using this approach for the 2930 cm$^{-1}$. See the picture below for typical peak areas from our samples.

[Figure]

While we certainly do not claim that we can completely eliminate mineral interference, we think that the specifically delimited 1620 cm$^{-1}$ peak that we use mostly consists of aromatic carbon i.e. the part of the peak that is selected is the part that disappears with the mentioned methods of SOC destruction. The finding, that it really is a meaningful proxy for carbon quality or stability is corroborated by the strong correlation (0.84) between the DSI and the percent of CPsoc, as was suggested to be computed by Lauric (new Figure S1 and comment below). As we recently demonstrated (Laub et al., 2019), and further found in the current study, the 2930 cm$^{-1}$ peak is also subject to interference even in non-carbonate containing soils. This is mostly by water, which can partly be removed by higher drying temperatures. So, in summary we believe that there is sufficient evidence that, even though there is noise in the DSI at both peaks, DSI is still a meaningful and useful proxy, which is highly correlated to other measures of SOC composition but has the advantage of being cost/time effective to measure.

To further illustrate how the 1620 cm-1 DRIFTS peak of bulk mineral soil is poorly related to SOM compounds, I provide the Figure 2 based on published and unpublished data from the paper of Barré et al. (2016) in Biogeochemistry showing the non-parametric Spearman's Rho coefficient of DRIFTS spectra from soils coming from the Ultuna Fame trial, one site that was used in this reviewed work by Laub and colleagues, with SOC concentration. In Figure 2, we clearly see the strong and positive Rho coefficient of the 2900 cm-1 spectral region with SOC concentration while the 1620 cm-1 spectral region show a Rho coefficient with SOC concentration close to 0, suggesting (though not demonstrating) that other compounds that organic matter absorb energy in the 1620 cm-1 spectral region of DRIFTS spectra, when scanning bulk mineral soils. From the above-mentioned information, I therefore question the rationale of the DRIFTS stability index of soil organic matter (SOM) in mineral soil samples.

The result is strongly affected by the delineation of the peak area. We thus agree if the whole 1620 cm-1 peak area (ca. 1755-1555 cm$^{-1}$) is taken results may not be reliable.

My interpretation is that this index is dividing a quantity that is highly correlated to SOC concentration (the 2900 cm-1 spectral region), by a quantity that is weakly changing when SOC concentration is modified (the 1620 cm-1 spectral region, provided a similar mineral composition). The DRIFTS stability index may thus show an increased SOC lability when SOC concentration is increased. I thus hypothesize that the DRIFTS stability index, as proposed by Demyan et al. (2012) and Laub and colleagues in this reviewed draft, may provide some information that is basically the same (though with added noise) than a variable much simpler than their index: total SOC concentration.

> We agree with the interpretation that the DSI is "dividing a quantity that is highly correlated to SOC concentration by a quantity that is weakly changing when SOC concentration is modified", and as we demonstrate above, both quantities are linked to forms of SOC. The fact that the selected subregion of the 1620 cm$^{-1}$ peak does not change strongly with SOC content, while, as destructive techniques demonstrate, it is still consisting mostly of aromatic carbon compounds (according to our integration limits), is exactly the reason why it is a very suitable proxy for slow turnover SOC.

It is well documented that an increase in SOC concentration is associated with an increased in the labile/stable SOC ratio, and all proposed indicators of SOM stability should be compared to SOC concentration, the most simple and straightforward indicator of SOM stability (though not very accurate).

What is the Spearman's Rho coefficient of the DRIFTS stability index with SOC concentration in the dataset of Laub and colleagues?

> We calculated the Pearson`s correlation coefficient -0.57 and Spearman`s rank correlation coefficient to be -0.68 between OC content and the DSI (as in formula 2) for the whole dataset (n=50). See the plot below

[Figure]

> We think, that the nonlinearity of the relationship between the DSI and the SOC content, as indicated by a higher rank correlation coefficient, points towards the possibility, that as SOC increases, most of the carbon is added to the fast turnover pool and that this could potentially be lost rather fast again.

I suggest that the authors (rather than using the spectral subtraction technique suggested by Nguyen et al., 1991 or Reeves, 2012), (i) test a soil dilution in KBr to reduce mineral artifacts in the 1620 cm-1 spectral region of neat DRIFTS, (ii) or test attenuated total reflectance mid-infrared spectroscopy (MIR-ATR) as an alternative technique.

Dilution with KBr (1:3 and 1:100) had been tested by Demyan et al. (2012), mainly to determine whether there was specular reflectance in the 1159 cm$^{-1}$ region, but was not found to yield better performances for deriving the DSI. Using neat samples avoids hygroscopic KBr which would have the potential to absorb water interfering with the 2930 cm$^{-1}$ and 1620 cm$^{-1}$ peaks and other non-desired interactions with the sample. We think that the major advantage of the DSI and DRIFTS PLSR is that it is possible to use undiluted bulk soil samples, and that it is nondestructive (cost effective and other analysis can be done with the same samples). We see the major advantage also in large scale applications, such as regional simulations, where other techniques are either too expensive or time consuming.

Indeed, MIR-ATR is a technique where the 1620 cm-1 peak region seems to be much less affected by quartz and other minerals that neat DRIFT signal, as illustrated in Figure 3 (Cécillon, Unpublished data).

From our understanding, the issue with ATR usually is that the signal throughput to the detector is weaker, thus the overall spectral features stand out less and are dominated by the silica vibrations at <1500 cm$^{-1}$, which is also shown in Figure 3 of Lauric Cécillon´s comment. The maximum absorbance in the DSI wavenumbers is almost an order of magnitude lower as compared to DRIFTS. If you zoom in on the figure, you can also see a small peak probably around 1620 cm$^{-1}$ in the silica sample, so it seems not to be free of mineral interference.

It might be possible that MIR-ATR is an alternative to DRIFTS, if it can reduce mineral interference at the 1620 cm$^{-1}$, but given the less strong signal of organic peaks it might be of limited use in low C soils. It could be worthwhile to do further research towards that direction and we think that this could be the content of another future publication.

Finally, as Laub and colleagues benefit from soil samples from two long-term bare fallow sites in Europe, I suggest that they compute the Spearman's Rho coefficient of their DRIFTS stability index with the proportion of centennially persistent soil organic carbon(CPsoc), that may be derived from the SOC evolution in the bare fallow plots, as shown by Cécillon et al. (2018).

A higher Spearman's rho coefficient of the DRIFTS stability index with CPsoc than the Spearman's rho coefficient of SOC concentration with CP-soc, would suggest an added value of the index compared to SOC concentration, in its current state.

Thanks for the discussion on this comment. We have now computed %CPsoc with the value of 6.95 g kg$^{-1}$ CPsoc from Ultuna derived by Cécillon et al. (2018) and 16.0 g kg$^{-1}$ CPsoc from Franko and Merbach (2017) for the bare fallow data we have available. As shown below, when combining the two datasets of Bad Lauchstädt and Ultuna the correlation between SOC and CPsoc across sites is poor. This shows that SOC alone is not a sufficient indicator for SOC quality. The correlation between the DSI and CPsoc on the other hand is quite strong (0.84), which according to Laurics comment is a strong indicator of its added value.

We think that it would be highly interesting to test this for other long-term bare fallows, where CPsoc could be mathematically derived (needing probably 30+ years of fallow) and this might help to optimize the DSI further. We think that a future publication could go into this direction and are excited about this finding.

[Figure]

As the reasoning behind CPsoc comes from RothC type models, which assume that there is only one actively decomposing SOC pool and another passive or inert SOC pool which is NOT subject to decomposition, this could mean that the DSI might also be useful for these types of models. In this study, we worked with DAISY, which is a CENTURY type model, that has a fast and slow SOC pool, both subjected to decomposition. We think that this is in agreement with the principle behind DRIFTS, and that microorganisms primarily target high energy aliphatic SOC, but aromatic SOC is also decomposed at a much slower rate, probably as a byproduct of enzyme release.

Overall, we very much acknowledge the issue of mineral interference addressed by the reviewer (see line 369 in the original manuscript) and the new addition:

information on SOM quality (Giacometti et al., 2013; Margenot et al., 2015).  While both peaks are subject to interference (2930 cm$^{-1}$ mainly from water and 1620 cm$^{-1}$ mainly from minerals (Nguyen et al., 1991)), it should be possible to limit the interference using subregions of the peaks with carefully selected

85  integration limits, only selecting the specific peak area of interest. Indeed, Demyan et al. (2012) found  aliphatic compounds to be enriched under long-term farmyard manure application and depleted in mineral fertilizer or control treatments, and showed that the  ratios of the 1620 cm$^{-1}$ to 2930 cm$^{-1}$ peak had a significant positive correlation with the

90  ratio of stable to labile SOM obtained by size and density fractionation. It was further corroborated that the specific integration limits of the peaks they used, which mainly selected the top subregion of the peak areas, are lost during combustion (Demyan et al., 2013). Hence, we hypothesized that the ratio of the aliphatic to aromatic DRIFTS

We have addressed this issue mainly by carefully delaminating the integration area and now have more clearly pointed to this in the methods:

peak areas of the four subsamples averaged after that. The local baselines were drawn between the intersection of the spectra and a vertical line at the integration limits (3010 − 2800 cm$^{-1}$ for the aliphatic C-H  stretching, 1660 − 1580 cm$^{-1}$ for aromatic C=C stretching vibrations). Example spectra and integrated peak areas are displayed in **Figure S 1**. These carefully selected integration limits were critical to reducing signal interference from water and minerals. Particularly, the mineral interference close to the 1620 cm$^{-1}$ peak makes accurate selection of integration limits necessary, so that only its top part (assumed to consist mostly of aromatic carbon) is selected. In the case of our samples, the selected specific peak area of the 1620 cm$^{-1}$ peak accounted for approximately 10 to 30 % of the total peak area (ca. 1755-1555 cm$^{-1}$), and roughly corresponds to the peak portion that is lost with combustion or chemical oxidation (Demyan et al., 2013; Yeasmin et al., 2017). A strong correlation between the DSI and the percentage of centennially persistent SOC (r = 0.84) from the combined long term experiments used in this study (using values of centennially persistent SOC from Cécillon et al., 2018; and Franko and Merbach, 2017), showed that the DSI selected in this manner did in fact explain a large portion of the SOC quality change across sites (**Figure S 2**).

We have further added a more detailed discussion on open questions of the DSI in the new manuscript version and finalize the section with the limitation that DSI should be tested before used with different soil types.

Compared with the other proxies for SOM quality discussed above, the measurements by DRIFTS are inexpensive, relatively simple, and the equipment of the same manufacturer is standardized. This should also constrain variability between different laboratories and be attractive for large-scale applications with large sample  numbers, for example to initialize simulations at the regional scale. However, for standardization of the DSI for model initialization one needs to address how the type of spectrometer (e.g. detector type) influences the spectra, if water and mineral interferences (Nguyen et al., 1991) in the spectra can be further reduced and if a mathematical standardization of the spectra and DSI (across instruments and water contents) is possible. While a complete elimination of mineral interference is not possible, a careful selection of integration limits and the use of a local baseline minimizes mineral interference of DRIFS spectra from bulk soils. This mostly selects the top part of the 1620 cm$^{-1}$ peak, which corresponds to the part that is reduced or completely lost when SOC is destroyed (Demyan et al., 2013; Yeasmin et al., 2017). Other approaches such as spectral subtraction of ashed samples or HF destruction of minerals prior DRIFTS analysis have been developed

in the attempt to obtain spectra of pure SOC. All are rather labor intensive and still produce artifacts, as it is not possible to destroy only the minerals or only the SOC without altering the respective other fraction (Yeasmin et al., 2017). Hence, we think that the selected integration limits might represent at this point the most feasible option for obtaining a robust and cost-effective proxy of SOC quality for modeling. The strong correlation of DSI and centennially persistent SOC as well as the model results of this study seem to corroborate this. The method of DSI estimation might be improved by a study of the best integration limits optimizing the fit of the DSI and centennially persistent SOC, which would require more bare fallow experiments than in this study. It could be worthwhile to use a purely mineral peak to correct for the mineral interference at 1620 cm$^{-1}$ similar to what was done to correct for carbonates in the 2930 cm$^{-1}$ peak by Mirzaeitalarposhti et al. (2016). The recent coupling of pyrolysis with

In the quest to find measurable fractions for model pools, we think that the DSI is a useful proxy (carefully selected integration limits, nonlinear relation with SOC, evidence that our 1620 cm$^{-1}$ is mostly from carbon, drying at 105 °C to reduce water interference at 2930 cm$^{-1}$). We first and foremost consider the DSI as a potential proxy to help initializing two pool SOM models, and our

question was, whether it was useful for this purpose or not, compared to steady state initializations. We think the value of this publication is to establish that the DSI has the potential to be a measurable fraction as a model pool proxy and thereby reducing model uncertainty, and show this to the scientific community. As any research this opens new questions which could lead to further development and refinement of the DSI. We think, that our study could demonstrate the DSI's usefulness and that it might be worthwhile to put further efforts and research towards its validation, use or optimization, especially because of its ease of use and inexpensive nature.

References:

[revised manuscript text omitted]

hat formatiert: Englisch (Vereinigte Staaten)

---

## Author Response (AR2)

Dear Editor, Dr. Michael Weintraub,

Dear reviewers,

We are happy to provide the revised version of the manuscript *bg-2019-292*. As a result of the reviewers' comments, we made some major changes. These include the way we address DRIFTS "peak areas" which are now correctly called areas below absorbance bands. We also corrected the wording for the 1620 cm$^{-1}$ band to aromatic/carboxylate which, due to higher specificity, we prefer to the suggestion of unsaturated carbon. In response to the comments of reviewer 4, we also clarified the concept of the DSI and how it is translated into an equation for SOM model pool partitioning. More details on the soil sampling and analysis were added, as was requested. We addressed all reviewers' comments and made the required changes based on these comments to the manuscript. We hope that with these recent changes, the manuscript now has the quality to be accepted for Biogeosciences.

We combined the responses to all authors in this PDF. Responses are in blue color, and a preview of changes made to the text in green color. The uploaded manuscript consists of a clean version, the version with tracked changes is attached at the end of this document.

We thank you for your effort in handling the manuscript during the review process.

On behalf of all coauthors,

Moritz Laub

**Lauric Cécillon:**

Dear authors, dear colleagues,

I thank you for your efforts to improve your revised draft and for testing new calculations (linking DSI and CPsoc). Incorporating such information as supplementary information is suitable to the readers of Biogeosciences.

I have one "technical", yet important comment on your revised draft.

In my view, naming as "aromatic C=C" the 1660-1580 cm-1 DRIFT region that is integrated around the 1620 cm-1 peak is not acceptable.

In my previous comments on your draft, I mostly emphasized on mineral artifacts in this DRIFT spectral region, so that the DSI is not carrying an information that is only linked to organic carbon.

But we know that C=O bounds are also absorbing energy on the 1660 cm-1 side of this DRIFT spectral region (as acknowledged by Demyan et al., 2012 in EJSS, https://doi.org/10.1111/j.1365-2389.2011.01420.x).
So beside mineral (+ water) artifacts, the 1660-1580 cm-1 DRIFT region represent a mixture of C=C + C=O carbon.
It thus not acceptable to name this region "aromatic carbon". This would make readers erroneously think that aromatic carbon is making the major part of stable carbon in soils, which is not true as stable carbon is generally richer in O than bulk SOM (except in soils rich in charcoals).

Thus I ask you to change the naming of the DRIFT 1660-1580 cm-1 spectral region to a proper (and scientifically correct) one such as "unsaturated carbon" (with C=O and C=C bounds ; see e.g. the Figure 1 of Pengerud et al., 2013 in Ecosystems, https://doi.org/10.1007/s10021-013-9652-5).

I am looking forward to reading the final and accepted version of your draft in Biogeosciences.

Kind regards,

Lauric Cécillon

Dear Lauric Cécillon,

Thank you for this very constructive and helpful suggestion. We agree that considering the involvement of C=O bonds in our defined 1620 cm$^{-1}$ absorbance band is appropriate, and hence changed the name of the absorbance band to aromatic/carboxylate throughout the manuscript. While we recognize that chemically the different functional groups we use are either unsaturated (C=C/C=O) or saturated (C-H), we feel that, specifying the vibrational assignments (e.g. aliphatic and aromatic/carboxylate) may avoid confusion with additional unsaturated functional groups in addition to C=C/C=O groups and provide clarity. Additionally, our band area of 1660-1580 cm$^{-1}$ does not correspond to the much larger band area of 1800 –1525 cm$^{-1}$, used by (Pengerud et al., 2013). As previously mentioned, we have clarified that the 1620 cm$^{-1}$ band includes both C=C and C=O functional groups.

Regards

**Reviewer 2:**

I criticized in my earlier review that Laub and colleagues use the chemical recalcitrance hypothesis to explain SOM cycling. Now, they cite an earlier study that their index, which is based on the aromatic and aliphatic peaks, is correlated with aggregation.

Mechanistically, the interpretation should be chemical recalcitrance. I don't get what we learn from DRIFTS spectra when we deviate from the mechanistic interpretation. Demyan et al. (2012) suggest a relation to mineral-association (correlation to the sum of the heavy fraction > 1.8 g cm-3 and clay fraction compared to light fraction < 1.8 g cm-3) and not to aggregation (the occluded light fraction is not measured in that study).

Arguably microaggregate occluded soil organic matter can be a part of the abovementioned heavy fraction > 1.8 g cm-3 and clay fraction but is possibly only a smaller portion (Castellano et al., 2015) of physicochemically stabilized SOC.

The sentences that Laub and colleagues added should reflect that the actual correlation is with physicochemically stabilized SOC (mainly mineral-association) and not aggregated SOC. Conceptually it is unclear for me why aliphatic SOC should not be preferentially found in the mineral-associated fraction since this should be mainly microbial residues.

We agree that this statement was oversimplified and not correct as it was. From our perspective, chemical recalcitrance is mostly related to attractiveness for microbial consumption, as all complexities of carbon can eventually be decomposed (Kallenbach et al., 2016). The sentence was removed and a larger discussion added.

> From a conceptual perspective DSI probably relates mainly to chemical recalcitrance of SOM present in different
> 515  SOM fractions. In that respect it is different from physical light/heavy fraction separation approaches as each of
> these fractions is very heterogeneous. For examples, tThe light fraction has strong absorbance at both
> aliphaticsaturated and aromatic/carboxylateunsaturated carbon bands (Calderón et al., 2011), so it could be that
> within each fraction, aliphatic saturated carbon is preferentially consumed by microorganisms. Thus, DSI reflects
> physicochemically stabilized SOC (mainly mineral-association in the case of bare soils) as also suggested by the
> 520  correlation of the ratio of 1620 cm⁻¹/2930 cm⁻¹ absorption bands areas to the ratio of mineral associated
> carbon/light fraction carbon (Demyan et al. 2012). The relationship to mineral-association is in many models is
> represented by a texture adjustment factor. On the other hand, DSI does not directly relate to aggregated (i.e.
> occluded) SOM, and its applicability in models focusing on aggregation needs to be evaluated (i.e. by a separate
> spectral analysis of occluded and remaining fractions.

I would appreciate if the authors could discuss this a little bit more also in light of the results from 13C-NMR which would suggest that the light fraction is enriched in aromatics (lignin) and the heavy fraction fractions in aliphatics (microbial cell walls). I think it might be worthwhile to discuss this further when I am missing it also other readers might.

We agree that these are interesting points, but this could be highly site specific and depend on the soil and residue type. From what we found in the literature the light fraction is not only enriched in aromatics but contains a strong band from aliphatic compounds as well (Calderón et al., 2011). From our perspective, using 13C-NMR as a gold standard also has it's drawbacks. The preparation requires completed demineralization of the soil, usually with HF acid. HF also destroys some of the SOM. Of course, it is then difficult to determine, which OM is destroyed. We hope that the paragraph we added to the discussion on this is to the liking of reviewer 2.

Apart from a more open discussion in this direction, I am happy with the changes made to the manuscript.

We are happy for this judgment of our article and would like to thank reviewer 2 for his constructive criticism during the review process of this paper. We think that the request for a more mechanistic understanding of the DSI helped to shape this article towards having a more process-oriented discussion.

Castellano, M.J., Mueller, K.E., Olk, D.C., Sawyer, J.E., Six, J., 2015. Integrating Plant Litter Quality, Soil Organic Matter Stabilization and the Carbon Saturation Concept. Global Change Biology.
Demyan, M.S., Rasche, F., Schulz, E., Breulmann, M., Müller, T., Cadisch, G., 2012. Use of specific peaks obtained by diffuse reflectance Fourier transform mid-infrared spectroscopy to study the composition of organic matter in a Haplic Chernozem. 63, 189-199.

**Reviewer 4:**

In general I like the idea of your study, however, (i) several mistaken terms need to be corrected ("vibrational peaks of absorbance", peaks instead of bands, "ratio of.. stretching vibrations"), (ii) the innovative aspect, and importance need to be clarified, and (iii) lots of details need to be clarified within material and methods (:most important is the DSI suggested to be a measure for SMB-C or SOM pools?). The paper need a carful revision (e.g. correction of mistakes terms, clarification of innovative aspect and importance, and addition of missing/ incomplete information in material and methods) before a complete review will be possible.

Some details:

There are some sematic mistakes:

a) the term "ratio of aliphatic C-H (2930 cm-1) to aromatic C=C (1620 cm-1) stretching vibrations" is not correct. According to textbooks on spectroscopy it should be called as a ratio between the area under aromatic and aliphatic absorption bands (which is also in accordance with the respective description in the materials and methods section in the manuscript under review.

Thank you for pointing out these terminology differences. We initially adapted the terms peak and peak area as is used in many publications (Egli et al., 2010; Pengerud et al., 2013; Stevenson, 1994; Yang, 2014). As this term seems to be suboptimal and is not used in specialized spectroscopy textbooks, we switched to the use of "area below absorption bands" as was suggested.

b) "vibrational peaks of absorbance (at L74 for example)". Should be called absorption bands: Please note that if infrared light of a certain wavelength introduces a vibrations at molecular scale that light will loss intensity which is recorded in the (FTIR/DRIFT) spectra by respective signals. However, since that light is absorbed by the sample such signals are called absorption bands (not peaks since this term is in general used to describe much sharper signals (textbooks on spectroscopy). Mistaken terms like the ones mentioned above need to be corrected. Standard terms need to be used. An introduction of new (incorrect / mistaken) ones need to be avoided.

We adapted the suggested terminology throughout the text.

c) In the abstract the "ratio of aliphatic C-H (2930 cm-1) to aromatic C=C (1620 cm-1)" which means A1620 to A2930 ratio is mentioned as a measure for DSI, but according to equation (2) it is the ratio between (A1620) and (A1620 +A2930), THIS is different! Please revise the text accordingly throughout the whole manuscript.

Thanks for this comment, we understand that this formulation could be confusing. We need to distinguish between the original definition of the DSI (A1620/A2930, Demyan eta al. 2012) and its corresponding use to partition SOM pools. We defined the DSI as the ratio of A2930/A1620 and set it to the ratio for SOM_fast/SOM_slow. Resulting from this, the fraction in the slow pool (analogous to the pool equation defined by Bruun and Jensen, 2002) is as the original equation: A1620/(A1620 +A2930). The fraction in the fast pool would be A2930/(A1620 +A2930). We have now both equations in the manuscript and added some text to explain this in detail.

> turnover rates and 49 % in the slow pool for the Bruun et al. (2003) turnover rates (**Table 2**). For the DSI
> initialization, the ratio of the area below the saturated absorption bands to the area below the unsaturated absorption
> band was used as the ratio of SOM in the fast to SOM in the slow cycling SOM pool:
>
> $$\quad \frac{\text{fast SOM}}{\text{slow SOM}} = \frac{A2930cm^{-1}}{A1620cm^{-1}} = DSI \tag{2}$$
>
> Thus, analogous to **Equation 1**, the fraction of SOM in the slow pool was calculated with the formula
>
> $$\text{slow SOM fraction} = \frac{A\_1620cm^{-1}}{A\_1620cm^{-1} + A\_2930cm^{-1}} \tag{32}$$
>
> With A_2930 cm$^{-1}$ and A_1620 cm$^{-1}$ being the specific  area  under the saturated and unsaturated carbon
> 250 band (described in section 2.1). The remaining carbon was allocated to the fast SOM

Abstract
- should consist of short statements on introduction, problem of interest, objective, material and methods, results, discussion and the conclusion. However, the problem of interest is missing.

We rewrote the introduction of the abstract and now clearly state the problem of model initialization.

L25-26. Please clarify the meaning of: "DRIFTS stability index ….., was used to divide SOM into fast and slow cycling pools in the soil organic module of the DAISY model." How was this done? How could a ratio be used to divide different pools? It may be usable to reflect the RATIO between SLOW and FAST pool or the proportion.

We now explicitly wrote, that 2930/1620 was set equal to fast/slow SOM, see comment to c).

Introduction
Clarify the innovative aspect and importance of your study.

We added two sentences to clarify.

1)

> losses that were to up to 30 % of initial SOM. Hence there is a need for a reproducible proxy for SOM pool
> initialization to reduce the high uncertainty of SOM models.

2)

> As a novel approach, this study uses information gained from DRIFTS spectra to partition measured SOM into
> pools of different complexity. DRIFTS can provide information on SOM quality, but also on texture and even

L73 Clarify the meaning of the sentence: "The interaction of mid-infrared energy with molecular bonds in soil produces typical vibrational peaks of absorbance at distinct wavelengths." It seems to be incomplete, refer to textbooks on FTIR/DRIFTs?

> et al., 2015; Tinti et al., 2015). The absorbance of mid-infrared light by molecular bonds
> in the soil sample vibrating at the same frequency  produce typical
> absorption bands at distinct wavelengths (Stevenson, 1994). The area below absorption bands (in short

L74 note the term "vibrational peaks of absorbance" is wrong. Absorption bands results from vibrations at the molecular level, but they are called in any textbook as absorption bands. All statement like the one above need to be changed throughout the whole manuscript..

*This was corrected throughout and addressed above in statement a)*

L74 Clarify the meaning of the term "These"

*We rewrote this section, see next comment.*

L75 This is incorrect, the wavelength of the CH absorption band is related to vibrations of C-H bonds in the SOM but NOT to elements. Please correct the wording and CLARIFY THE MEANING:

absorption bands  (Stevenson, 1994).  area below absorption bands (in short  can be linked to different molecular bonds of carbo, amides, silicates

L77 Stretching peak need to be corrected into absorption band thoughout the whole manuscript…..

*This was corrected and addressed above in statement a)*

L75 to 80 please clarify and consider that this problem is already described by textbooks on spectroscopy but also by authors (e.g. Capriel 1997, Celi et al. 1997 among others) who interpret the CH absorption bond with respect to soil wettability. These authors already mentioned the overlap of CH bands by those of OH bands. BUT they also stated that the OH bands are not only present in water the OH is also part of SOM, and soil minerals (e.g. clay minerals, oxides, hydroxides, etc.). These OH band will remain in air dried soil samples as well in the ones dried at 37 to 105°C. But drying at higher temperature may affect the OH content of SOM and soil minerals thereby affecting the absorption bands in DRIFT/FTIR spectra (see Yeasmin, Reeves among others). Please correct this statement AND clarify how you ensure that the drying procedure did not affect the spectral properties of your samples. Add respective spectra into the supplement.

*There is some evidence that the spectral properties of samples does not change after drying at 105°C (Duboc et al., 2016) as there were no significant differences between spectra of samples dried at different temperatures/methods when stored outside a desiccator. What we found, is that when drying at 105°C the 2930 cm$^{-1}$ band area has the best visibility as the surrounding OH band absorption is reduced. We observed that leaving samples outside the desiccator for several hours led to a regain of OH band absorption, but we did not specifically measure this effect, as this was not our focus. We used different drying temperatures for deriving the DSI for the initial test of the DSI because we were not sure, if high temperatures would have negative effects for the utility of the DSI for modelling. Spectra and more detailed explanation below.*

L86 explain the background of the idea of SOM pool initialization.
*This is explained in detail in the first paragraph of the introduction.*

Material and Methods:
PLEASE describe in more detail the soil sampling. Which soils were taken where, at which time? Which is the year of establishment for each of the experiments, at which year did the bare-fallow started, and which are the sampling year(s)?

*We added the season of sampling for Ultuna and Bad Lauchstädt, the frequency of measurements for the Kraichgau and Swabian Jura sites and wrote explicitly that experiment and bare fallow establishment were in the same years. For those interested in the exact sampling dates, we added the raw data of this study into the supplementary which includes dates of sampling.*

L145 ff seem mostly belong to the discussion.

This detailed explanation of the integration process was added within the review process, as the clarification of this was a major request of the other reviewers.

How did you ensure that drying at 105°C did not result in artifacts? Add references different to your own work.

We tested the different temperatures to explore this issue. In our prior work (Laub et al., 2019) we found that there is a linear decrease with drying temperature of the broad OH band surrounding 2930 cm$^{-1}$, which goes along with a linear increase of absorption bands area related to carbon, which mainly comes from a reduction of clay OH signal. This smooth trend with temperature does from our perspective not point towards artifacts. Still we were not sure, whether 105°C is the best representation of organic absorption bands, which is the reason why we tested different temperatures for modeling, respective SOM pool initializations. Our reasoning for the modelling was as follows: If drying produces artifacts, then model performance should be reduced, but it improved. We can also refer to the work of Duboc et al. (2016), who found no differences in ATR spectra between freeze drying, 60 °C and 105°C when samples were not stored in a desiccator. We attach example spectra of different drying temperatures as well as the (3x scaled) difference between the absorbance at 105°C and 32°C (32-105 °C) to illustrate the development of absorption bands. As can be seen, there is a gain of absorbance with higher drying temperatures for the most part of the spectra, while absorbance from the broad OH band is reduced (positive difference). From our perspective, drying at 105 °C therefore makes the SOM related bands better visible due to the removal of interfering OH. Additionally, using a combined in situ thermal DRIFTS and evolved gas technique (Demyan et al., 2013), soil samples were heated incrementally from 25 to 700 °C and either the evolved gas quantified or the mid-infrared spectrum of the soil itself was measured. $CO_2$ evolution, which would imply combustion or off-gassing, was not found at temperatures up to 105 °C. As the overall goal is to get the best representation of SOM within the DRIFTS spectra, 105 °C seems the most appropriate drying temperature for our application.

[Figure]

Figure 1 Baseline corrected example DRIFTS Spectra of bulk soil from the Swabian Jura (Table S1), dried at different temperatures. The red line indicates the difference of the 32 °C–105 °C absorbance (3× scaled for better illustration). The integration limits of the peak areas of interest are shown by vertical lines.

Add the limits of the band areas, how did you chose them?

The limits are stated (L143 of the prior version of the manuscript). They were initially chosen by Demyan et al. 2012; 2013, attempting to best resolve the 2930 and 1620 $cm^{-1}$ bands. This was based on isolated soil fractions, as well as spectra from substances rich in aromatic compounds, such as xylan and tannic acid. For the 2930 $cm^{-1}$ band, the range 3010-2800 $cm^{-1}$ was chosen as this avoids the aromatic C-H vibrations (if present) found around 3030 $cm^{-1}$ (Stevenson, 1994) and in soils investigated the 2930 $cm^{-1}$ aliphatic contribution (often a doublet on the very wide OH band) terminates at 2800 $cm^{-1}$. For the 1620 $cm^{-1}$, the limits of 1660-1580 $cm^{-1}$ were chosen as to include the anticipated range of aromatic C=C vibration variations while avoiding the amide band at <1580 $cm^{-1}$ and C=) of COOH and ketones at >1660 $cm^{-1}$ and significant mineral interference (e.g. Demyan et al., 2013)We fully recognize that including wavenumbers between 1630-1660 $cm^{-1}$ will probably also include C=O stretching of H-bonded C=O (Bellamy, 1975; Stevenson, 1994), thus have added in the manuscript (which was also cited in the Demyan et al., 2012 paper) that there is also a contribution of C=O within the band limits we use .

L171: meaning of the term "DAISY pools"? The ones mentioned in table 2 but table 2 show turnover rates, "maint" and "CUE"?

These additional parameters (maintenance respiration and carbon use efficiency) are explained in detail in Mueller et al. (1997). They are normal parameters in SOM models, but we did not alter them. We included them for completeness of the parameter set but decided for the main text to only include the relevant parameters that were changed and are relevant for the message, as readers can go back to original publications. We added a sentence to the caption of figure 2, to explain the difference between microbial turnover and death.

Describe the DSI evaluation, - initialization….

Those are described in chapter 2.3 and 2.4.

L194: Where is the description of SMB-C measurement? Please add a short s of SMB-C determination.

A description of SMB-C exists already at L 155 ff in the manuscript.

Please explain your reasons for dividing SMB-C into a fast and a slow cycling microbial pool. Why are they important?

This division is a default of DAISY. We added to the text, that we did a standard division, as this division is not the focus of this study.

According to the statements at L194 to 203 the SMB-C seemed to be assumed here as a measure for SOM. Is this correct, please explain! Please clarify the differences between the SMB-C pools (L194) and the SOM pools (L199)! AND CLARIFY which pool ratios will be described when using the A1620/(A1620+A2930) ratio?

We clarified in the text, that as bulk soil SOC included carbon from SMB-C, this needs to be subtracted from the amount of SOC that makes up the SOM pools carbon.

> Measured bulk soil SOC includes SMB-C, therefore the amount of SOC in the fast and slow cycling SOM pools was bulk soil SOC minus SMB-C. As not focus of this study, a standard division of Mmeasured SMB-C was divided into the slow and fast cycling microbial pools was done, with 10 % in the fast (8 % in Mueller et al., 1998) and 90 % in the slow pool. The remaining carbon (difference between total bulk soil SOC and SMB-C) was divided
> 225  between fast and slow cycling SOM pools either by either the DRIFTS stability index (DSI), or according to the

We also added a more detailed explanation of the DAISY pool to the section, where the models are explained.

> and Linux systems. The DAISY model consists of two pools (fast and slow cycling) for each of the measurable fractions of 1) litter, 2) SMB and 3) stabilized SOM (**Figure 1**). Due to bare fallow, litter pools were disregarded in this study and the focus was on initializing the two SOM pools. A detailed description of the DAISY SOM

L405 please add background information for this statement (table, Fig.?, ref).
Did you consider somewhere in the discussion that the Bad Lauchstädt soil is a Chernozem? Please ad missing information, old carbon present in Chernozem may cause difference in SOM turn-over (papers on black carbon etc…).

We added the reference to the figure 3, altered the text and added a new reference to this.

> turnover rates. In the case of the Chernozem of Bad Lauchstädt, only turnover rates had an influence on model performance and. The its SOC turnover properties of the Chernozem were was generally not well captured overestimated with by either both parameter sets (**Figure 3**). It was previously suggested that the high SOC storage capacity of this site is related to a high content of adsorbed cations (Ellerbrock and Gerke, 2018). , and it probably
> 440  has a slower overall SOM decomposition than many other agricultural soils.So a possible reason for an overestimation of SOC turnover in Bad Lauchstädt might be, that DAISY only considers clay content as stabilizing mechanism. Nevertheless, the use of DSI also was suitable for Bad Lauchstädt, as it did not reduce model

I stopped reading here since lots of information need to be clarified and added before a careful review will be possible.

We are grateful for the critical comments of the reviewer, who seems to be an expert in the field of SOM and soil spectroscopy. We think that addressing these, had further improved the quality of the manuscript.

Table 1 please add the studied sites, AND note SOC stand for SOIL organic carbon content please correct the capture accordingly refer to standard captures…
Explain the terms - UTM (abbreviation for ?), -"Depth of measurement" possibly the depth of sampling?, - the meaning of initial (SOC in soil in the year the bare fallow is established, of the SOC of soil in the year the experiment was established?
Please add year of establishment and sampling; number of replicates

We added experiment name and reference for Ultuna and Bad Lauchstädt, year of establishment abbreviation of UTM, changed to depth of sampling and that fallow establishment and experiment establishment were at the same time.

Table 2 which of the pools are assumed to represent the fast/slow cycling SMB-C pools mentioned at L194?

We added additional explanation for the SMB pools which are subject to maintenance respiration and microbial death at the same time. We noticed that the SMB pool was wrongly named in figure 1 and changed this. The pools names are SMB and SOM – they contain C and N, but we did not focus on N simulation.

Table 3 show starting and ending year of the field experiment as well as … SOC etc … of the soil sampled at….. site. Please correct accordingly.

We added a longer heading, to explain the meaning of experiment start and simulation end etc. Correct the capture of table 4 and 5 accordingly…

We assessed and corrected the Table 4 and 5 once more.

Fig. 1 explain the meaning of the term "A_XXXX"

Was done.

Fig. 2 I assume the capture should be: baseline corrected and vector normalized DRIFT spectra of 105°C dried soil samples from soils under bare fallow at … site. Please indicate which figure shows what kind of spectrum. How did you ensure that drying at 105°C did not result in artifacts? If these spectra are baseline correct WHY did they start at different Y-axis values?

We changed the caption as requested. In the caption it is stated that the Y-axis offset was merely added for a better visibility. We added the size of offset (+0.02 for samples from the start).

References for the responses:

Bruun, S. and Jensen, L. S.: Initialisation of the soil organic matter pools of the Daisy model, Ecol. Modell., 153(3), 291–295, doi:10.17665/1676-4285.20155108, 2002.

Calderón, F. J., Reeves, J. B., Collins, H. P. and Paul, E. A.: Chemical Differences in Soil Organic Matter Fractions Determined by Diffuse-Reflectance Mid-Infrared Spectroscopy, Soil Sci. Soc. Am. J., 75(2), 568, doi:10.2136/sssaj2009.0375, 2011.

Demyan, M. S., Rasche, F., Schulz, E., Breulmann, M., Müller, T. and Cadisch, G.: Use of specific peaks obtained by diffuse reflectance Fourier transform mid-infrared spectroscopy to study the composition of organic matter in a Haplic Chernozem, Eur. J. Soil Sci., 63(2), 189–199, doi:10.1111/j.1365-2389.2011.01420.x, 2012.

Demyan, M. S., Rasche, F., Schütt, M., Smirnova, N., Schulz, E. and Cadisch, G.: Combining a coupled FTIR-EGA system and in situ DRIFTS for studying soil organic matter in arable soils, Biogeosciences, 10(5), 2897–2913, doi:10.5194/bg-10-2897-2013, 2013.

Duboc, O., Tintner, J., Zehetner, F. and Smidt, E.: Does sample drying temperature affect the molecular characteristics of organic matter in soil and litter? A statistical proof using ATR infrared spectra, Vib. Spectrosc., 85, 215–221, doi:10.1016/j.vibspec.2016.05.002, 2016.

Egli, M., Mavris, C., Mirabella, A. and Giaccai, D.: Soil organic matter formation along a chronosequence in the Morteratsch proglacial area (Upper Engadine, Switzerland), CATENA, 82(2), 61–69, doi:10.1016/j.catena.2010.05.001, 2010.

Kallenbach, C. M., Frey, S. D. and Grandy, A. S.: Direct evidence for microbial-derived soil organic matter formation and its ecophysiological controls, Nat. Commun., 7, 1–10, doi:10.1038/ncomms13630, 2016.

Laub, M., Blagodatsky, S., Nkwain, Y. F. and Cadisch, G.: Soil sample drying temperature affects specific organic mid-DRIFTS peaks and quality indices, Geoderma, 355, 113897, doi:10.1016/j.geoderma.2019.113897, 2019.

Mueller, T., Jensen, L. S. S., Magid, J. and Nielsen, N. E. E.: Temporal variation of C and N turnover in soil after oilseed rape straw incorporation in the field: simulations with the soil-plant-atmosphere model DAISY, Ecol. Modell., 99(2), 247–262, doi:http://dx.doi.org/10.1016/S0304-3800(97)01959-5, 1997.

Pengerud, A., Cécillon, L., Johnsen, L. K., Rasse, D. P. and Strand, L. T.: Permafrost Distribution Drives Soil Organic Matter Stability in a Subarctic Palsa Peatland, Ecosystems, 16(6), 934–947, doi:10.1007/s10021-013-9652-5, 2013.

Stevenson, F. J.: Humus chemistry: genesis, composition, reactions, John Wiley & Sons, New York., 1994.

Yang, X.: An extension to "Mid-infrared spectral interpretation of soils: Is it practical or accurate?," Geoderma, 226–227(1), 415–417, doi:10.1016/j.geoderma.2014.03.022, 2014.

[revised manuscript text omitted]

SOM = soil organic matter pools,

[revised manuscript text omitted]